# Alternative splicing of PBRM1 mediates resistance to PD-1 blockade therapy in renal cancer

Namjoon Cho [ID][1], Seung-Yeon Kim[1], Sung-Gwon Lee[2], Chungoo Park [ID][2], Sunkyung Choi [ID][3✉],
Eun-Mi Kim [ID][4✉] & Kee K Kim [ID][1✉]

## Abstract

Alternative pre-mRNA splicing (AS) is a biological process that results in proteomic diversity. However, implications of AS alterations in cancer remain poorly understood. Herein, we performed a comprehensive AS analysis in cancer driver gene transcripts across fifteen cancer types and found global alterations in inclusion rates of the PBAF SWI/SNF chromatin remodeling complex subunit *Polybromo 1* (*PBRM1*) exon 27 (E27) in most types of cancer tissues compared with those in normal tissues. Further analysis confirmed that *PBRM1* E27 is excluded by the direct binding of RBFOX2 to intronic UGCAUG elements. In addition, the E27-included PBRM1 isoform upregulated PD-L1 expression via enhanced PBAF complex recruitment to the *PD-L1* promoter. *PBRM1* wild-type patients with clear cell renal cell carcinoma were resistant to PD-1 blockade therapy when they expressed low *RBFOX2* mRNA levels. Overall, our study suggests targeting of RBFOX2-mediated AS of *PBRM1* as a potential therapeutic strategy for immune checkpoint blockade.

**Keywords** Alternative Splicing; Immune Checkpoint Inhibitor; PBRM1; PD-L1; RBFOX2
**Subject Categories** Cancer; Immunology; Translation & Protein Quality

## Introduction

Alternative pre-mRNA splicing (AS) is a process that produces multiple mRNA variants via differential selection of exons in the pre-mRNA of a single gene, thereby expanding proteomic diversity (Bludau and Aebersold, 2020). During the evolution of eukaryotes, AS was greatly diversified, contributing to the complexity of organisms (Barbosa-Morais et al, 2012). In humans, more than 95% multiple-exon genes generate various splicing isoforms via AS (Pan et al, 2008). Alternative exons have relatively weak splice-site consensus sequences (Merkin et al, 2012). Therefore, the inclusion or exclusion of the exons is strongly determined by the direct pre-mRNA binding of splicing factors that contribute to or interfere with spliceosome formation (Van Nostrand et al, 2020b). AS events programmed to determine cell fate contain highly conserved *cis*-regulatory elements and are regulated via systemic changes in the expression and activation of *trans*-acting splicing factors in response to various cellular processes, such as signal transduction, differentiation, proliferation, and survival (Choi et al, 2023; Choi et al, 2022; Mazin et al, 2021; Paronetto et al, 2016).

Abnormal AS regulation plays a crucial role in organ dysfunction and pathogenesis of human diseases, including cancer (Daguenet et al, 2015; Scotti and Swanson, 2016). Recent studies have identified alterations in global splicing pools in cancer tissues via comprehensively analysing AS signatures (Chan et al, 2022; Kahles et al, 2018). These studies provide important information about patterns of splicing abnormalities in cancer tissues or the impact of AS alterations in some cancer patients who harbour somatic mutations in specific splicing factors, including *U2AF1* and *SF3B1* (Kahles et al, 2018; Zhang et al, 2022). However, the identification of cancer-associated AS alterations remains challenging owing to the vast number of genes with dynamic AS changes in cancer tissues and lack of information about functional differences between splicing isoforms. Therefore, there is a need for approaches to identify oncogenic AS alterations in cancer tissues for developing therapeutic targets and diagnostic strategies.

In this study, to simplify the complexity of AS analysis, we focused on AS alterations within cancer driver genes that play a role in cancer development by promoting oncogenic activities or diminishing tumour suppressive functions due to somatic mutations (Bailey et al, 2018). The aberrant transcriptional state of cancer driver genes due to mutations or amplification is a crucial risk factor for cancer development and progression (Chen et al, 2014). This indicates that altered function of cancer driver genes plays an important role in cancer development. Although AS is an important process that results in functional changes in proteins by forming various isoforms, the specific impact of AS alterations in cancer driver genes in cancer tissues is less explored.

Here, we conducted a comprehensive analysis of AS alterations in cancer driver genes across the 15 cancer types. Our analysis revealed AS alterations of *Polybromo-1* (*PBRM1*) exon 27 (E27) in

[1]Department of Biochemistry, College of Natural Sciences, Chungnam National University, Daejeon 34134, Republic of Korea. [2]School of Biological Science and Technology, Chonnam National University, Gwangju 61186, Republic of Korea. [3]Department of Biological Sciences, College of Natural Sciences, Keimyung University, Daegu 42601, Republic of Korea. [4]Department of Bio & Environmental Technology, College of Science and Convergence Technology, Seoul Women's University, Seoul 01797, Republic of Korea. ✉E-mail: skchoi@kmu.ac.kr; eunmi.kim@swu.ac.kr; kimkk@cnu.ac.kr

most types of cancer tissues. Furthermore, we elucidated the regulatory mechanism of AS in *PBRM1* and its impact on clinical outcomes in patients with cancer. Overall, the results herein provide new insights into diagnostic and therapeutic strategies to improve the survival of patients with cancer.

# Results

## AS alterations of *PBRM1* in cancer

To compile a list of key cancer driver genes, we ranked the top 60 most frequently mutated somatic genes in 15 cancer types from The Cancer Genome Atlas (TCGA) (Appendix Fig. S1A). We then analysed the alterations in AS events in these genes between normal and cancer tissues using TCGA SpliceSeq database (Appendix Fig. S1B,C). Our analysis revealed 104 AS events in 36 cancer driver genes. To identify the AS events that affect multiple types of cancer, we reclassified the ones detected in at least 12 of the 15 cancer types, based on their percent spliced in index (PSI; Ψ) values (Fig. 1A). Next, we calculated the mean change in PSI value in pan-cancer tissues compared with that in normal tissues and found that *PBRM1* E27 exhibited the highest mean |ΔPSI| value. Subsequently, we selected eight genes (*PBRM1*, *HERC2*, *KRAS*, *NF1*, *RNF213*, *NSD1*, *TRRAP*, and *CHD4*) with the highest average |ΔPSI| scores, except for *FN1* where AS site specification was difficult, and analysed the proportion of patients with cancer having AS changes in these genes by comparing a cancer tissue with an adjacent normal tissue derived from the same patient registered with TCGA database (Fig. 1B). The results showed that alterations in |ΔPSI| values of *PBRM1* E27, by more than 0.15, occurred in over 49% patients with cancer, which was the highest frequency among AS events. Therefore, we focused on the AS events of *PBRM1* E27.

*PBRM1*, a tumour suppressor gene comprising 30 exons, encodes a subunit of the polybromo-associated BAF (PBAF), a subtype of the mammalian switch/sucrose non-fermentable (mSWI/SNF) complex (Mashtalir et al, 2018) (Fig. 1C). In 11 of 15 cancer types, *PBRM1* E27 was significantly more included in cancer tissues than in normal tissues, whereas clear cell renal cell carcinoma in TCGA (KIRC) showed E27 exclusion (Fig. 1D). We also found that *PBRM1* mRNA levels showed no obvious differences between normal and cancer tissues across TCGA cancer types. Splice variant analysis of TCGA data showed that E26 and E27 are cassette exons of *PBRM1*; however, we did not observe any significant difference in the PSI values of *PBRM1* E26 between normal and cancer tissues (Appendix Fig. S2A), leading us to focus only on E27 AS events in *PBRM1*. To validate TCGA results, PCR was performed using cDNA arrays of human cancer tissues with a primer pair targeting the E25 and E28 of *PBRM1* (Appendix Fig. S2B). Next, we performed agarose gel electrophoresis to assess the AS pattern of *PBRM1*. Due to the similar size of *PBRM1* E26 and E27, comprising 165 and 156 nucleotides, respectively, we further extracted the DNA from the middle range of the PCR product-containing agarose gel bands, amplifying the E26+ E27− and E26− E27+ variants. Next, we mixed the samples with Apa1 restriction enzyme, specifically cutting the *PBRM1* E27 region, to identify the proportions of the *PBRM1* splice variants (Appendix Fig. S2C). Endometrial cancer cDNA array analysis showed that E27 inclusion in *PBRM1* was significantly higher in cancer tissues than in normal

tissues (Fig. 1E). Similarly, breast cancer cDNA array analysis revealed a significant inclusion pattern of *PBRM1* E27 in cancer tissues compared with that in normal tissues (Fig. 1F). These findings supported the results of TCGA analysis and demonstrated the *PBRM1* E27 AS patterns are significantly altered in most cancer tissue types compared with that in normal tissues.

## RBFOX2 promotes E27 exclusion by directly binding to *PBRM1* pre-mRNA

To investigate the regulatory mechanism of *PBRM1* E27 AS, we analysed the correlation between PSI values of *PBRM1* E27 and expression levels of 172 splicing factors in TCGA cancer datasets (Figs. 2A and EV1A). Our analysis revealed a strong negative correlation between *PBRM1* E27 PSI and RBFOX2 expression in most cancer types. Heatmaps of TCGA data for individual patients demonstrated a robust inverse relationship between *PBRM1* E27 PSI and *RBFOX2* mRNA expression (Figs. 2B and EV1B). In addition, cancer types with inclusion patterns of *PBRM1* E27 showed a downregulated pattern of *RBFOX2* expression in cancer tissues compared with that in normal tissues (Figs. 2C and EV1C). Based on these profiles, we hypothesised that RBFOX2 is a key regulator of *PBRM1* E27 AS. Given that both E26 and E27 are cassette exons, we designed primers to amplify each splice variant separately (Fig. 2D). We used an E22 and E23 targeting forward and reverse primer F1 and R1, respectively, to amplify the constitutive exons of the *PBRM1* transcripts and assess total *PBRM1* mRNA levels. To specifically amplify E26 or E27 included splice variants, we used the E25 and E26 targeting forward and reverse primers F2 and R2 or the E27 and E28 targeting forward and reverse primers F3 and R3, respectively. Finally, we used the forward and reverse primers F2 and R3, respectively, to amplify all the *PBRM1* splice variants, regardless of whether E26 and E27 were included or excluded in the *PBRM1* transcripts, resulting in the amplification of the *PBRM1* E26− E27− splice variant as a lower-DNA size-product, quantified after agarose gel electrophoresis. We knocked down *RBFOX2* in U-2 OS cells, which showed a higher RBFOX2 protein level than that in HeLa cells, and over-expressed RBFOX2 in HeLa cells (Fig. 2E). *RBFOX2* depletion induced E27 inclusion in U-2 OS cells (Fig. 2F) and that RBFOX2 overexpression reduced E27 inclusion in HeLa cells in a dose-dependent manner (Fig. 2G). However, changes in RBFOX2 expression did not affect the AS patterns of E26 in *PBRM1* or total *PBRM1* mRNA expression (Fig. EV2A,B). In addition, altering the E27 AS did not influence the total PBRM1 protein level. We further confirmed the RBFOX2-mediated *PBRM1* E27 AS regulation in HEK293 and MCF7 cells (Fig. EV2C,D).

Next, we assessed how RBFOX2 induces the exclusion of *PBRM1* E27. RBFOX2 enhanced crosslinking and immunoprecipitation (eCLIP) sequencing data from ENCODE portal showed RBFOX2 protein binding in the upstream intronic region of *PBRM1* E27 (Fig. 3A) (Consortium, 2012; Van Nostrand et al, 2020a). RBFOX2 has a distinct RNA binding sequence: 5′-UGCAUG-3′ (Lovci et al, 2013). We found three UGCAUG sequences in the intronic region upstream of human *PBRM1* E27 (Fig. 3B). Using the UCSC PhyloP conservation score to examine the degree of sequence conservation, we confirmed that all three UGCAUG sequences were highly conserved across vertebrates. Since RBFOX2 is known to bind to UGCAUG elements and

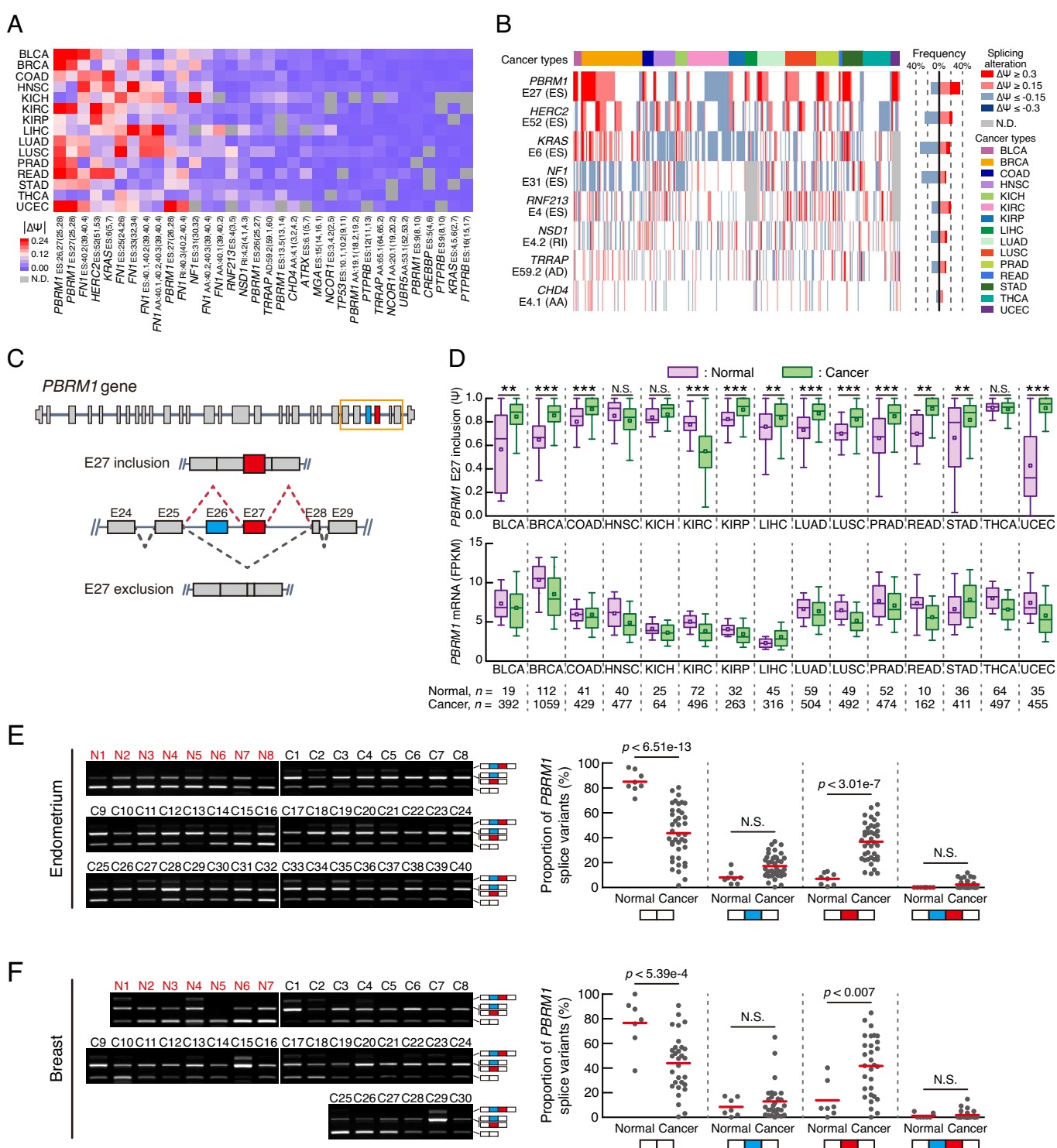

regulate the AS of target exons, we further investigated whether *PBRM1* E27 AS regulation by RBFOX2 was mediated by its binding to UGCAUG elements present in the upstream intronic region of *PBRM1* E27. We first investigated the interaction between RBFOX2 protein and *PBRM1* pre-mRNA using an RNA immunoprecipitation (RIP) assay with an anti-RBFOX2 antibody (Fig. 3C). Using primers that detect the flanking regions of the three UGCAUG

elements, we confirmed that RBFOX2 bound to *PBRM1* pre-mRNA. Next, we investigated the functional roles of the three UGCAUG elements using minigene constructs containing E27, flanking E25 and E28, as well as *PBRM1* introns (Fig. 3D). Transfection of the *PBRM1* minigene constructs into U-2 OS cells showed that when any one of the three UGCAUG elements was mutated, the E27 exclusion of *PBRM1* minigene transcripts was

**Figure 1.   A comprehensive pan-cancer AS analysis reveals extensive changes in the AS pattern of *PBRM1* E27 in cancer tissues.**

(A) Heatmap showing the AS events of cancer driver genes across 15 cancer types. The colour indicates the change in mean PSI value in cancer tissues compared with that in normal tissues. The bottom of the heatmap shows the gene name, AS type, splicing event exon, upstream exon, and downstream exon. Only the events with quantified PSI values in ≥80% of the 15 cancer types are represented in the heatmap. Ψ, PSI; ES, exon skipping; AA, alternative acceptor; AD, alternative donor; RI, retained intron; N.D., not detected. (B) Frequency of splicing alteration for selected genes from (A). The coloured line in the heatmap (left) indicates the patient with differentially spliced events between cancer tissues and adjacent normal tissues. The bar graph (right) shows the proportion of splicing alternations in each AS event. (C) Schematic representation of the AS of human *PBRM1*. (D) The AS patterns of *PBRM1* E27 (top) or the mRNA levels of *PBRM1* (bottom) in TCGA. Boxes represent the median, quartiles, 10th percentile, and 90th percentile. *n*, number of samples. *p*-values were calculated using the two-tailed Student's *t*-test. (BLCA, *p* < 0.002; BRCA, *p* < 7.12e−26; COAD, *p* < 0.001; KIRC, *p* < 1.63e−33; KIRP, *p* < 1.62e−5; LIHC, *p* < 0.006; LUAD, *p* < 1.78e−10; LUSC, *p* < 1.76e−9; PRAD, *p* < 1.16e−8; READ, *p* < 0.002; STAD, *p* < 0.009; UCEC, *p* < 4.57e−11). (E, F) RT-PCR analyses of *PBRM1* AS in human endometrium (E) and breast (F) samples including cancer (C) and healthy (N) tissues. The *PBRM1* splice variant proportions in each tissue is presented in the dot plots (right), with red lines indicating the mean. Schematics at the bottom of the graphs represent *PBRM1* splice variants. White boxes, E25 and E28; Blue box, E26; Red box, E27. *p*-values were calculated using one-way ANOVA with Bonferroni's multiple comparison test. N.S., not significant. Source data are available online for this figure.

reduced (Fig. 3E). In particular, the mutant (Mut) 2 construct showed a greater reduction in E27 exclusion than the wild-type (WT), Mut 1, and Mut 3 constructs. *RBFOX2* knockdown in U-2 OS cells induced E27 inclusion in almost all the transcripts of the minigene constructs, with or without the UGCAUG site mutation. We further observed E27 exclusion of the *PBRM1* minigene transcripts by RBFOX2 overexpression in a UGCAUG element-dependent manner in HeLa cells (Fig. 3F). In addition, using a *PBRM1* minigene construct containing another mutation sequence in the UGCAUG sites, we validated the effects of UGCAUG elements on the E27 AS patterns of the *PBRM1* minigene transcripts (Fig. EV2E–G). Our findings demonstrated that all three UGCAUG elements influenced RBFOX2-mediated E27 exclusion, with the second UGCAUG element being the most crucial. We performed an RNA pull-down assay using the biotinylated upstream intron of *PBRM1* E27 with the RBFOX2 protein obtained from in vitro transcription and translation under cell-free conditions (TnT) (Fig. 3G). Results indicated that RBFOX2 bound directly to the UGCAUG site without the assistance of other proteins. Taken together, our results demonstrated that RBFOX2 directly binds to the highly conserved UGCAUG elements in the upstream intron of *PBRM1* E27, leading to the exclusion of *PBRM1* E27.

## E27-included PBRM1 upregulates PD-L1 expression

We investigated the functional differences across the PBRM1 splicing isoforms. First, we established HeLa cell lines expressing the E27-excluded isoform of *PBRM1* (ΔE27) using the CRISPR-Cas9 system (Figs. 4A and EV3A). Immunoblot analysis confirmed that there was no difference in PBRM1 protein levels in ΔE27 cells compared with those in WT cells. In addition, our RT-PCR analysis confirmed that the AS patterns of *PBRM1* E26 were not altered in the ΔE27 cells compared to WT cells (Fig. 4A). To analyse the cellular effects of E27 exclusion in *PBRM1*, we performed RNA sequencing using WT and ΔE27 cells and conducted gene set enrichment analysis (GSEA) with hallmark gene sets from the global mRNA expression data (Figs. 4B and EV3B). Results revealed a significant downregulation of interferon (IFN)-responsive gene signature in ΔE27 cells (Figs. 4C and EV3C). STRING protein interaction analysis using the top 120 downregulated genes in ΔE27 cells also indicated a strong enrichment in IFN signalling-associated annotations (Fig. EV3D). Using qRT-PCR, we validated the downregulation of several IFN-responsive genes, including

programmed death ligand-1 (*PD-L1*) mRNA, in ΔE27 cells relative to those in WT cells (Fig. 4D). PD-L1, upregulated by IFN gamma (IFNγ) and oncogenic signalling in cancer cells, plays a crucial role in inhibiting the killing activity of immune cells targeting cancer cells (Garcia-Diaz et al, 2017; Yamaguchi et al, 2022). Therefore, we investigated the role of *PBRM1* E27 in *PD-L1* expression. Treatments of WT and ΔE27 cells with epithelial growth factor (EGF) and hepatocyte growth factor (HGF), to stimulate oncogenic signalling, showed that *PD-L1* mRNA levels were decreased in ΔE27 HeLa cells compared with those in WT cells, even when *PD-L1* expression was induced by treatment with EGF or HGF (Fig. 4E,F). To investigate whether the difference in *PD-L1* expression, depending on AS pattern of *PBRM1* E27, was due to the induction activity of the E27-included isoform or the inhibitory activity of the E27-excluded isoform, we transfected siPBRM1 into WT and ΔE27 HeLa cells and treated the latter with EGF and IFNγ to induce PD-L1 protein expression (Figs. 4G and EV3E,F). Immunoblot and qRT-PCR analyses showed that *PBRM1* knockdown in WT HeLa cells reduced the mRNA and protein levels of PD-L1. In contrast, *PBRM1* knockdown in ΔE27 HeLa cells did not affect PD-L1 expression levels. Moreover, we revealed that knockdown of E27-included isoform using siRNA targeting PBRM1 E27 sequences repressed PD-L1 expression in WT HeLa and MDA-MB-231 cells, but not in ΔE27 HeLa cells (Fig. EV3G,H). These findings suggested that the E27-included isoform induced PD-L1 expression, whereas the E27-excluded isoform did not affect PD-L1 expression. Next, we conducted a dual-luciferase assay using a *PD-L1* promoter reporter vector and observed reduced *PD-L1* promoter activities in the ΔE27 cells compared to the WT cells (Fig. EV3I). However, these changes remained markedly smaller than the endogenous PD-L1 expression level differences between the WT and ΔE27 cells, suggesting that E27-included PBRM1 may regulate *PD-L1* expression through epigenetic mechanisms. Overall, these results indicate that the inclusion of E27 in PBRM1 enhances PD-L1 expression in cancer cells.

## Differential binding activity of PBRM1 isoforms on the *PD-L1* promoter

PBRM1 is a subunit of the PBAF complex that affects the local binding of PBAF complex to the target gene regions (Sinha et al, 2020; Yao et al, 2023). To investigate how the E27-included isoform of PBRM1 promotes PD-L1 expression, we performed co-immunoprecipitation (Co-IP) using an anti-PBRM1 antibody to

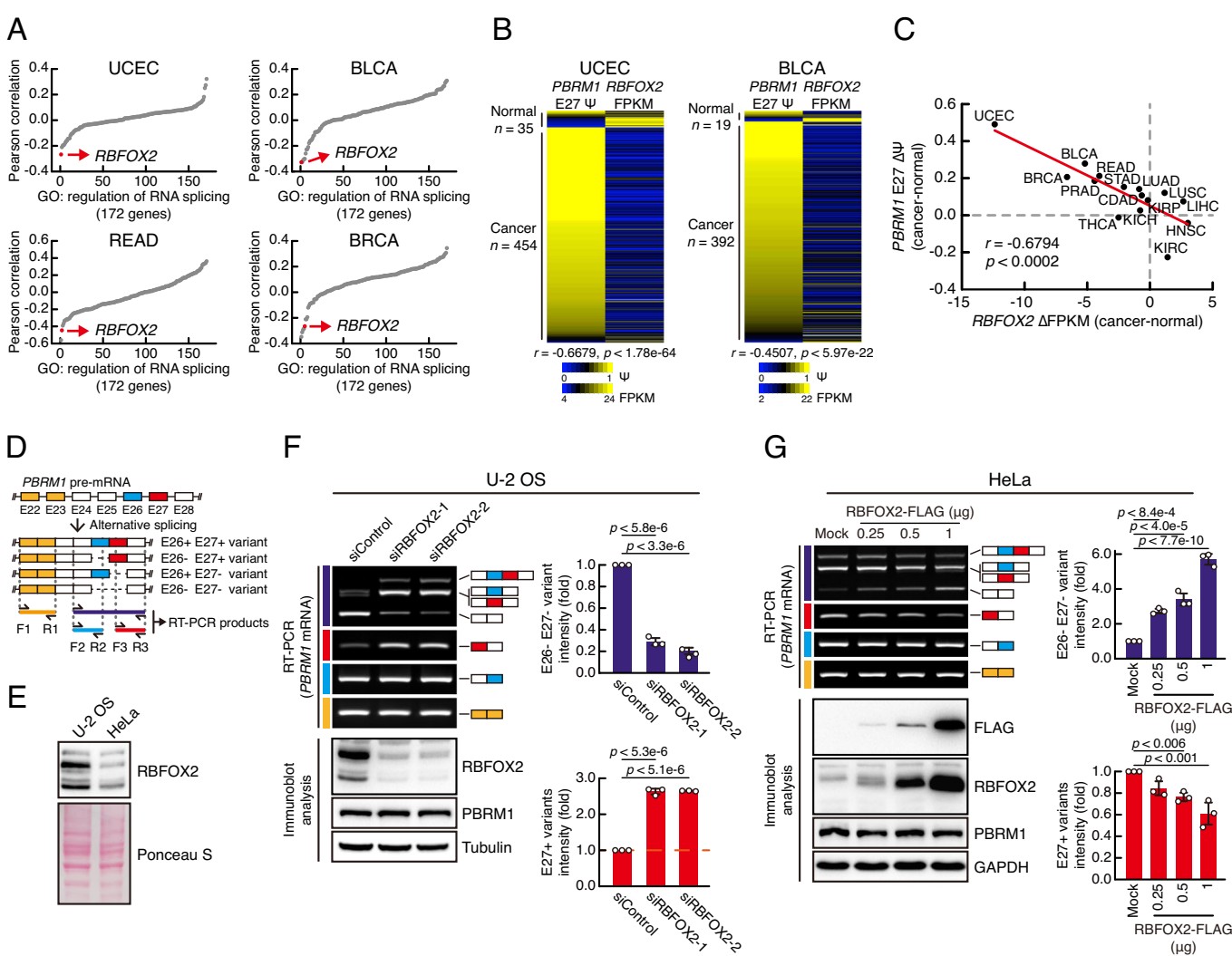

**Figure 2. RBFOX2 regulates the E27 AS of *PBRM1*.**

(A) Dot plots displaying splicing factors according to the correlation between mRNA levels of splicing factors and PSI values of *PBRM1* E27 in cancer tissues. (B) Heatmaps of individual UCEC and BLCA samples showing an inverse correlation between the PSI values of *PBRM1* E27 and mRNA levels of *RBFOX2*. Pearson correlation coefficient and *p*-value are presented. *n*, number of samples; Ψ, PSI. (C) Dot plot of the changes in *PBRM1* E27 AS versus changes in *RBFOX2* expression in different cancer types compared to those in normal tissues. Pearson correlation coefficient and *p*-value are presented. (D) Schematic representation of *PBRM1* splice variants and its RT-PCR products. (E) Immunoblot analysis of RBFOX2 in U-2 OS and HeLa cells. (F, G) RT-PCR (top) and immunoblot (bottom) analyses showing the expression of *PBRM1* splice variants in *RBFOX2*-knockdown U-2 OS cells (F) or in RBFOX2-FLAG-overexpressed HeLa cells (G). Coloured lines on the left of the RT-PCR results indicate the RT-PCR products, as explained in (D). Quantification of RT-PCR products for E27+ variants and E26− E27− variant are shown in bar graphs (right) (*n* = 3, biological replicates). Bars indicate mean ± SD. *p*-values were calculated using one-way ANOVA with Dunnett's multiple comparison test. Source data are available online for this figure.

confirm the interaction of PBRM1 splicing isoforms with the PBAF complex subunits (Fig. 5A). The assembly of PBRM1 splicing isoforms in the PBAF complex remained unaffected, irrespective of whether the E27 was included or excluded in PBRM1 protein. Co-IP of the ARID2 protein, a core subunit of the PBAF complex, also confirmed that *PBRM1* E27 AS did not affect PBAF complex assembly (Fig. 5B). Moreover, there was no significant difference in subcellular localisation across the PBRM1 splicing isoforms (Fig. EV3J). Therefore, we analysed the binding of PBRM1 splicing isoforms to the *PD-L1* promoter region. We performed the chromatin immunoprecipitation (ChIP) assay using an anti-PBRM1 antibody in WT and ΔE27 HeLa cells, followed by qRT-

PCR analysis using primers targeting the *PD-L1* promoter region; results revealed that the E27-excluded isoform exhibited weaker binding activity towards the *PD-L1* promoter than the E27-included isoform (Fig. 5C–E). To verify whether the increased binding activity of the PBRM1 splicing isoform due to E27 inclusion affects the binding activity of the PBAF complex, we performed ChIP of the ARID2 protein in WT and ΔE27 HeLa cells and found that ARID2 binding to the *PD-L1* promoter region was significantly reduced in ΔE27 HeLa cells; therefore, the E27-included isoform of PBRM1 might guide the PBAF complex to bind to the *PD-L1* promoter region, thereby inducing PD-L1 upregulation.

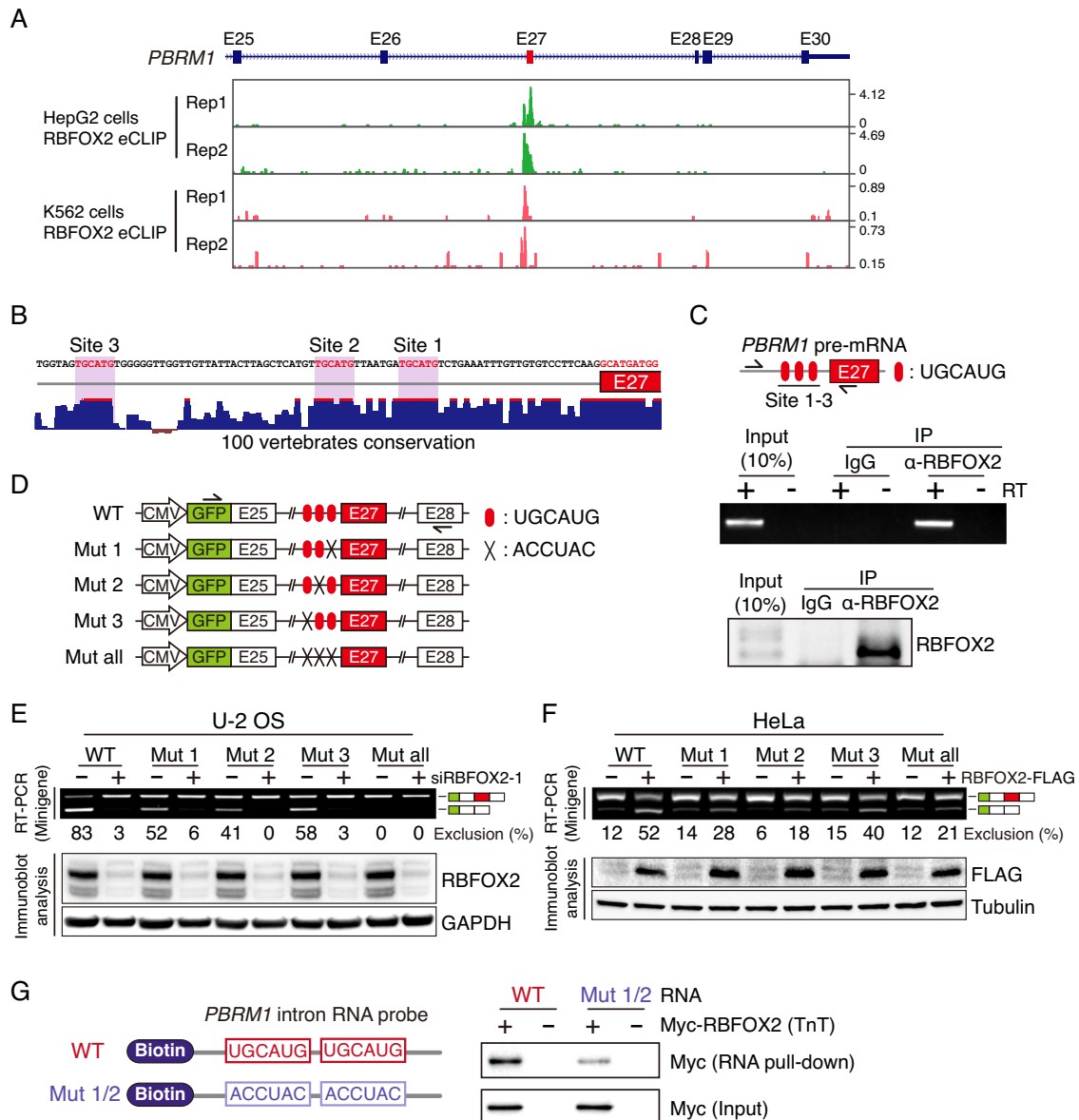

**Figure 3. RBFOX2 directly binds to the UGCAUG elements in *PBRM1* pre-mRNA.**

(A) Genome browser view displaying the RBFOX2 eCLIP peaks on the *PBRM1* transcripts. (B) Conserved UGCAUG elements in the upstream intronic region of *PBRM1* E27 across 100 vertebrates. (C) RIP assay showing the interaction of RBFOX2 with the *PBRM1* pre-mRNA. Schematic (top) representation of the primer-binding site on the *PBRM1* pre-mRNA for RIP-PCR. RT-PCR (middle) and immunoblot (bottom) analyses were performed following the RIP assay. (D) Schematic representation of *PBRM1* minigene constructs. (E, F) RT-PCR (top) and immunoblot (bottom) analyses for the AS pattern of the *PBRM1* minigene transcripts by RBFOX2 knockdown in U-2 OS cells (E) or by RBFOX2-FLAG overexpression in HeLa cells (F). (G) RNA pull-down showing the direct binding of RBFOX2 to a biotinylated *PBRM1* intron RNA probe. Source data are available online for this figure.

## Association of *PBRM1* E27 AS with tumour progression

Next, we hypothesised that the induction of E27 exclusion in *PBRM1* enhances immune cell-mediated tumour-killing activity, leading to tumour suppression. We first co-cultured NK-92 cells with WT or ΔE27 HeLa cells, and observed that ΔE27 HeLa cells were more sensitised to NK-92 cell-mediated tumour-killing activity compared to WT HeLa cells (Fig. EV4A). We next synthesised a PBRM1 splice-switch oligonucleotide (SSO) to induce E27 exclusion in PBRM1

(Fig. EV4B). Transfection of PBRM1 SSO into various cancer cell lines not only promoted E27 exclusion in *PBRM1* but also reduced cancer cell resistance to NK-92 cell-mediated tumour-killing activity (Fig. EV4C). Finally, we transfected the PBRM1 SSO into B16-F10 cells and injected them subcutaneously into the backs of mice (Fig. EV4D). At 9 days post-injection, the E27 exclusion in *Pbrm1* by PBRM1 SSO transfection significantly reduced the tumour volume (Fig. EV4E,F). Taken together, the results suggest that the E27 inclusion in PBRM1 contributes to tumour progression.

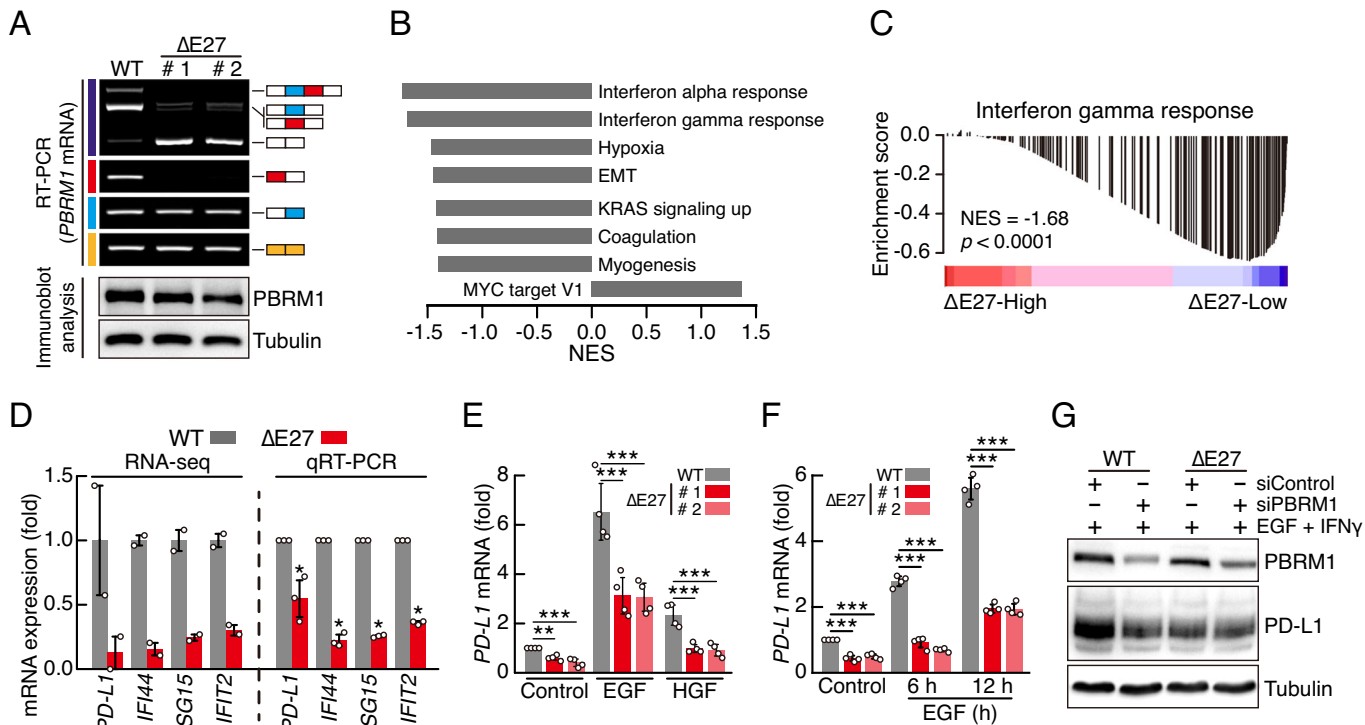

**Figure 4.   E27 inclusion of PBRM1 promotes PD-L1 expression.**

(A) RT-PCR (top) and immunoblot (bottom) analyses showing the establishment of ΔE27 HeLa cell lines. (B) GSEA of RNA sequencing data for all gene expression in ΔE27 HeLa cells versus WT HeLa cells. Significantly enriched terms at FDR < 0.0025 and nominal p < 0.01 are represented in the bar graph. (C) GSEA of IFNγ-responsive genes in all gene transcripts of ΔE27 HeLa cells versus WT HeLa cells. The nominal p-value is presented. (D) Validation of RNA sequencing data for IFN-responsive genes by qRT-PCR analysis (n = 3, biological replicates). Bars indicate mean ± SD. p-values were calculated using two-tailed Student's t-test. (PD-L1, p < 0.047; IFI44, p < 0.002; ISG15, p < 6.86e−5; IFIT2, p < 0.001). (E) qRT-PCR analysis of PD-L1 mRNA levels in 20 ng/ml EGF- or 40 ng/ml HGF-treated ΔE27 HeLa cells for 12 h (n = 4, biological replicates). Bars indicate mean ± SD. p-values were calculated using one-way ANOVA with Dunnett's multiple comparison test. (Control: WT vs # 1, p < 0.004; WT vs # 2, p < 0.001; EGF: WT vs # 1, p < 1.54e−4; WT vs # 2, p < 1.40e−4; HGF: WT vs # 1, p < 2.49e−4; WT vs # 2, p < 1.88e−4). (F) qRT-PCR analysis of time-dependent PD-L1 mRNA levels in HeLa cells treated with 20 ng/ml EGF (n = 4, biological replicates). Bars indicate mean ± SD. p-values were calculated using one-way ANOVA with Dunnett's multiple comparison test. (Control: WT vs # 1, p < 8.83e−6; WT vs # 2, p < 1.68e−5; 6 h: WT vs # 1, p < 1.84e−6; WT vs # 2, p < 9.39e−7; 12 h: WT vs # 1, p < 5.26e−8; WT vs # 2, p < 5.12e−8). (G) Immunoblot analysis of PD-L1 in 20 ng/ml EGF- and 100 ng/ml IFNγ-treated HeLa cells for 12 h following transfection with siPBRM1 for 36 h. Source data are available online for this figure.

## Clinical outcomes of patients with cancer

The identification of AS alterations of *PBRM1* E27 in cancer and its role in cancer immune evasion raises questions about their impact on patients with cancer. Thus, we further analysed TCGA database and focused on patients with KIRC, which only showed widely distributed PSI values of *PBRM1* E27 in cancer tissues between 15 cancer types (Fig. 6A). Approximately, 40% patients with ccRCC harboured *PBRM1* mutations. Therefore, we classified patients with KIRC into the *PBRM1* WT, *PBRM1* Mut, and *PBRM1* low Fragments Per Kilobase of exon per Million fragments mapped (FPKM) groups; the latter possessed the non-mutant *PBRM1* gene with low *PBRM1* mRNA levels (Fig. EV5A,B). We analysed *PD-L1* mRNA levels based on the PSI value of *PBRM1* E27 and found that, in the *PBRM1* WT group, patients with high PSI values of *PBRM1* E27 had significantly higher levels of *PD-L1* mRNA expression than those with low PSI values of *PBRM1* E27 (Fig. 6B,C). However, in the *PBRM1* Mut group, there was no significant difference in *PD-L1* mRNA levels between patients with high and low PSI values of *PBRM1* E27. PSI values of *PBRM1* E27 and mRNA levels of *PD-L1* were significantly correlated in the *PBRM1* WT group but not in

the *PBRM1* Mut group (Fig. EV5C). Notably, the *RBFOX2* expression was negatively correlated with *PBRM1* E27 PSI, regardless of the presence of the *PBRM1* mutation (Figs. 6D and EV5D). These results suggest that the E27-included isoform of PBRM1 is required for *PD-L1* upregulation, while the regulatory system of *PBRM1* E27 AS, mediated by the RBFOX2 expression level, does not directly affect *PD-L1* mRNA levels.

Perforin and granzyme B are expressed and secreted by immune cells, such as cytotoxic T cells and natural killer (NK) cells, to activate cancer cell death. However, when PD-L1 is expressed in cancer cells and interacts with PD-1 receptor in immune cells, it can inhibit the immune response, leading to a reduction in perforin and granzyme B expression and secretion. Therefore, we hypothesised that *PD-L1* expression, according to the PBRM1 expression status, could affect perforin and granzyme B mRNA levels by inhibiting the tumour-killing activity of immune cells. Further analysis revealed that perforin and granzyme B mRNA levels were significantly decreased in *PBRM1* WT patients with high PSI values of *PBRM1* E27 compared to those in other *PBRM1* WT patients (Fig. 6E,F). As expected, *PBRM1* Mut patients showed no significant differences in perforin and granzyme B mRNA levels

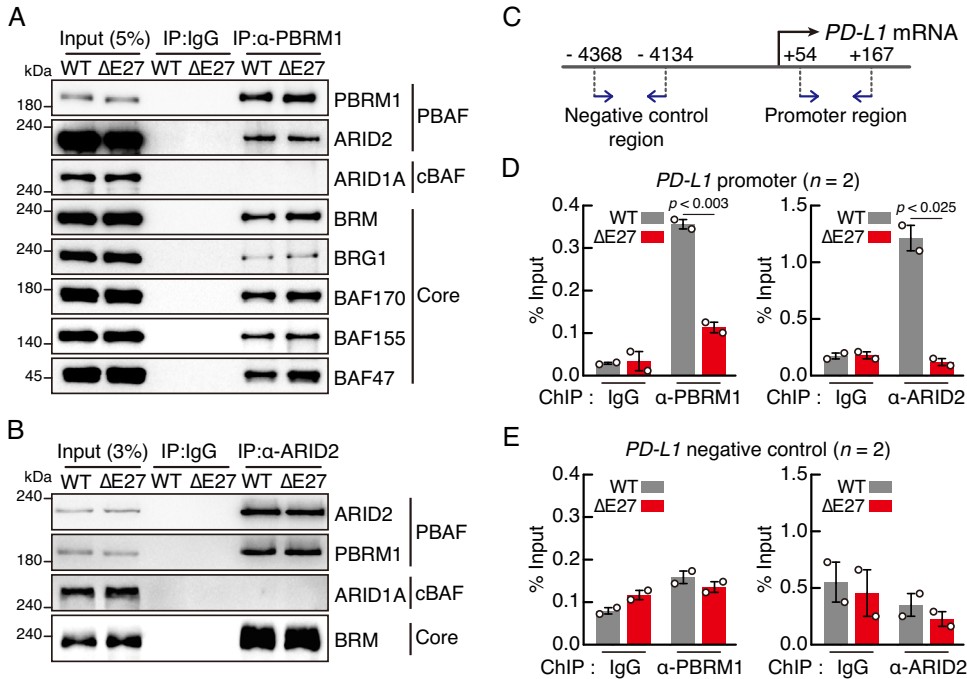

**Figure 5. The E27-included isoform of PBRM1 increases PBAF binding to the *PD-L1* promoter region.**

(A, B) IP with an anti-PBRM1 antibody (A) or an anti-ARID2 antibody (B) in WT or ΔE27 HeLa cells followed by immunoblot analysis. PBAF, polybromo-associated BAF; cBAF, canonical BAF; Core, pan-mSWI/SNF complex subunit. (C) Schematic representation of *PD-L1* gene. The blue arrows indicate primer-binding sites. (D, E) ChIP-qPCR amplifying the *PD-L1* promoter region (D) or negative control region (E) from an anti-PBRM1 antibody (left) or an anti-ARID2 antibody (right)-precipitated immune complexes from HeLa cell lysates ($n = 2$, biological replicates). Bars indicate mean ± SD. *p*-values were calculated using one-tailed Student's *t*-test. Source data are available online for this figure.

between the groups with high and low PSI values of *PBRM1* E27. In addition, mutations in *VHL* genes, which are the most frequently mutated in ccRCC, did not affect *PD-L1* mRNA levels in cancer tissues (Fig. 6G). Overall, the results suggested that the E27-included isoform of PBRM1 promotes *PD-L1* expression in ccRCC cancer tissues and suppresses the tumour-killing activity of immune cells.

We next analysed whether the AS pattern of *PBRM1* E27 affects clinical outcomes of cancer patients treated with immune checkpoint inhibitors. Nivolumab (anti-PD-1) has been approved by Food and Drug Administration (FDA) for the treatment of ccRCC. Notably, a recent study analysing patients with ccRCC enrolled in nivolumab clinical trials revealed that patients with a mutation in *PBRM1* showed improved survival rates after nivolumab treatment. Therefore, we further determined the impact of *PBRM1* E27 AS on the overall survival of patients with ccRCC who were treated with PD-1 blockade therapy. For analysing samples showing effects due to the expression of *PBRM1* splicing isoforms, patients with ccRCC possessing the non-mutant *PBRM1* gene and a low *PBRM1* mRNA level (below 12%) were not classified in the *PBRM1* WT group. In the *PBRM1* WT group, patients with low *RBFOX2* mRNA levels had lower overall survival rates than patients with high *RBFOX2* mRNA levels (Fig. 6H). In contrast, patients with a mutation in *PBRM1* showed no significant difference in overall survival rates between the groups with high and low *RBFOX2* mRNA levels. Moreover, overall survival rates of patients with ccRCC, who were treated with everolimus (mTOR inhibitor), showed no significant difference based on *RBFOX2*

mRNA levels in the *PBRM1* WT group (Fig. 6I). In addition, mutations in *VHL* genes did not affect overall survival rates of patients with ccRCC, regardless of whether *RBFOX2* mRNA levels were high or low (Fig. EV5E,F). Next, we analysed the overall survival rates of nivolumab-treated patients with ccRCC according to their *PD-L1* mRNA levels. The results revealed no significant difference in overall survival rates according to *PD-L1* expression status in nivolumab-treated patients with ccRCC, although the patients enrolled in clinical trials exhibited a higher proportion of high *PD-L1* mRNA levels in the high E27 functional group (*RBFOX2* low and *PBRM1* WT) compared to the low E27 functional group (all others) (Fig. EV5G,H). This result suggests that the impact of *PBRM1* E27 AS on the PD-1 blockade therapy resistance might be influenced by variable factors beyond the *PD-L1* expression levels. Collectively, these findings suggested that patients with ccRCC, who harbour WT *PBRM1* and express *PBRM1* mRNA with a high rate of E27 inclusion, show high resistance to PD-1 blockade therapy.

# Discussion

In this study, a comprehensive analysis of AS in key cancer driver genes uncovered splicing alterations of *PBRM1* E27 in most cancer types. Analysis of AS events is essential for understanding the detailed mechanisms of biological processes, including the pathogenesis of human diseases. In the present study, we explored cancer from a new perspective based on AS alterations. Given the

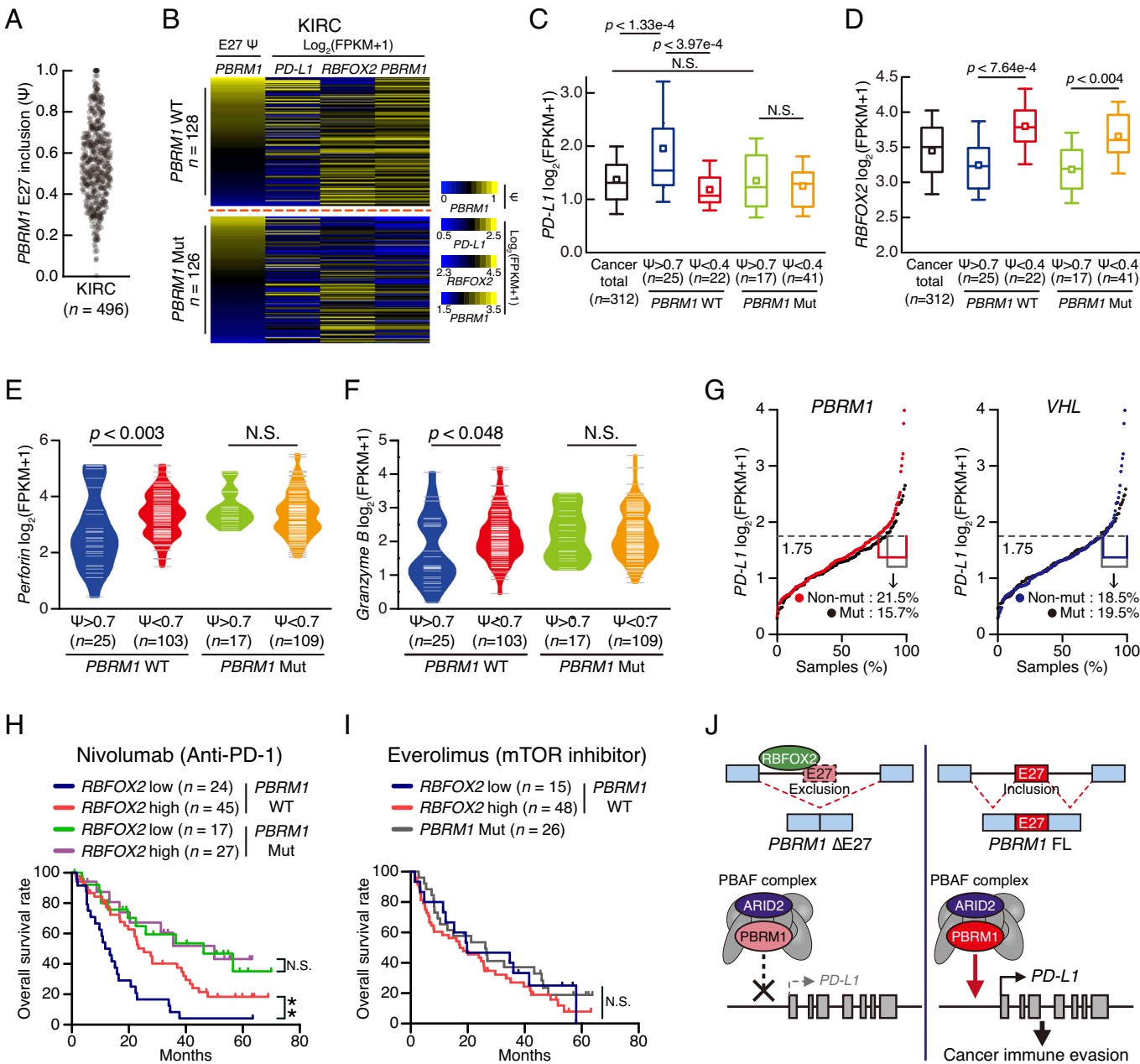

**Figure 6.  RBFOX2-mediated AS in *PBRM1* affects resistance to PD-1 blockade therapy in ccRCC.**

(A) Dot plot showing PSI values of *PBRM1* E27 in individual KIRC cancer tissues. *n*, number of samples; Ψ, PSI. (B) Heatmap showing the correlation between PSI values of *PBRM1* E27 and mRNA levels of *PD-L1*, *RBFOX2*, and *PBRM1* in individual KIRC tissues. *n*, number of samples. (C, D) Box plots showing the *PD-L1* (C) or *RBFOX2* (D) mRNA levels according to PSI values of *PBRM1* E27 in the *PBRM1* WT KIRC group or *PBRM1* Mut KIRC group. Boxes represent the median, quartiles, 10th percentile, and 90th percentile. *p*-values were calculated using one-way ANOVA with Bonferroni's multiple comparison test. *n*, number of samples; N.S., not significant. (E, F) Violin plots showing the mRNA levels of perforin (E) or granzyme B (F) according to PSI values of *PBRM1* E27 in patients with KIRC. *p*-values were calculated using the two-tailed Student's *t*-test. *n*, number of samples; N.S., not significant. (G) Dot plot showing individual KIRC patients with percent of samples expressing *PD-L1* more than 1.75 values of normalised FPKM depending on mutation in *PBRM1* (left) or *VHL* (right). (H, I) Kaplan–Meier survival curves of patients with ccRCC treated with nivolumab (H) or everolimus (I) from clinical trial data (Checkmate 009, 010, and 025). *p*-values were calculated using Log-rank test. **$p < 0.002$. *n*, number of samples; N.S., not significant. (J) Model for RBFOX2-mediated regulation of E27 AS in *PBRM1* and its role in cancer immune evasion. Source data are available online for this figure.

widespread influence of AS on human gene transcripts and extensive alterations of AS events in cancer cells, we focused on AS events within cancer driver genes. We believe that this approach was advantageous for identifying key AS alterations associated with cancer progression. Using this approach, we discovered the prevalent splicing alterations of *PBRM1* E27 in cancer tissues. Moreover, we found cancer type-specific alterations of AS patterns in cancer driver genes, such as *HERC2*, *KRAS*, and *NF1*. Based on these results, it can be speculated that cancer driver genes potentially influence cancer development and progression

depending on their AS patterns in cancer tissues. Therefore, further research on AS alterations in cancer driver genes will be crucial for understanding their detailed roles in cancer.

The RBFOX family (RBFOX1, 2, and 3) proteins play a crucial role in organ development, including brain, heart, and muscle development, by regulating global AS patterns (Jacko et al, 2018b; Kim et al, 2013; Singh et al, 2018; Singh et al, 2014a; Wei et al, 2015b). RBFOX1 and RBFOX3 are predominantly expressed only in neuronal cells, whereas RBFOX2 is expressed widely in various tissues and various cell types, including stem cells (Kim et al, 2011; Underwood et al, 2005). RBFOX2 is pivotal for embryonic stem cells survival as well as brain, heart, and muscle development (Jacko et al, 2018a; Singh et al, 2014b; Wei et al, 2015a; Yeo et al, 2009). However, recent studies have shown an association between RBFOX2 functions and cancer progression (Choi et al, 2022; Choi et al, 2019; Jbara et al, 2023; Park et al, 2017). RBFOX proteins possess an RNA-recognition motif that predominantly binds to UGCAUG elements, thereby regulating the AS pattern of exons adjacent to their binding sites (Conboy, 2017; Ye et al, 2023). Here, we observed that RBFOX2 binds to the upstream intronic region of E27 and promotes its exclusion in *PBRM1* pre-mRNA. Notably, we found a high level of conservation in all three UGCAUG elements. Based on these findings, we speculate that systemic and dynamic alterations in the E27 inclusion rate in *PBRM1*, depending on RBFOX expression levels, may play pivotal roles in the embryonic stem cell maintenance, developmental stages, and disease progression across vertebrates.

PBRM1 harbours six bromodomains that bind to modified histone proteins, two bromo-adjacent domains, and a high mobility group (HMG) box that may bind to DNA; however, the detailed roles of these components have not been studied (Karki et al, 2021; Liao et al, 2019; Wang et al, 2022b). The HMG-box consists of three α-helices, and we discovered that both E26 and E27 of *PBRM1* encode the third α-helix of the HMG-box. Secondary structure predictions confirmed that, regardless of whether the E26 or E27 is included or excluded, the third α-helical structure in the HMG-box region is encoded by *PBRM1* splice variants (Appendix Fig. S3). This suggests that AS-mediated modification of the HMG-box in PBRM1 potentially influences its interactions with target DNA. We also revealed differential binding activities on the *PD-L1* promoter region between PBRM1 splicing isoforms, depending on the E27 inclusion, and it affected the ARID2 binding activity on the *PD-L1* promoter region (Fig. 6J). These findings suggest that RBFOX-mediated systemic regulation of *PBRM1* AS affects PBAF complex-mediated genome-wide accessibility and global gene expression levels. Additional investigations are needed to determine the specific alterations of PBAF complex functions according to AS alterations of PBRM1 or other mSWI/SNF family subunits in various cell types, including cells in cancer tissues.

We reported differential survival rates, according to the AS pattern of *PBRM1* E27 and the mutation status of *PBRM1*, in patients with ccRCC treated with nivolumab but not with everolimus. Based on our findings, we speculated that patients with ccRCC harbouring the WT *PBRM1* gene and a high E27 inclusion in the *PBRM1* pre-mRNA may expect greater therapeutic effects from everolimus treatment than from nivolumab treatment. Thus, our results provide an important standard for screening patients with ccRCC, who are potentially resistant to immune checkpoint inhibitor therapy and may benefit from alternative treatment options. Such insights could be valuable for developing precision medicine approaches to increase the lifespan of patients with cancer.

The role of PBRM1 in the resistance of cancer cells to immune cells remains debatable. Several studies, from CRISPR-Cas9 screening to patient-cohort studies, have demonstrated that the loss of function in PBRM1 reduces the resistance of cancer to immune checkpoint inhibitor therapy (Braun et al, 2020; McDermott et al, 2018; Miao et al, 2018; Pan et al, 2018). However, consistent roles of *PBRM1* loss-of-function mutations in patients with cancer or in cancer cell lines have not been determined (Hakimi et al, 2020; Liu et al, 2020; Wang et al, 2022a). Our study revealed that the PBRM1 expression level and *PBRM1* mutation alone cannot fully explain the functional roles of PBRM1 in cancer. Furthermore, we showed that AS of *PBRM1* E27 is a critical factor in clinical outcomes of PD-1 blockade therapy in patients with ccRCC. Therefore, our findings provide an important key to resolve discrepancies in PBRM1 function reported in previous studies and shed light on the importance of AS patterns of *PBRM1* E27 in cancer.

Here, although we focused on the impact of *PBRM1* E27 AS on patients with ccRCC treated with nivolumab, we also uncovered the diverse frequency of E27 inclusion rates in *PBRM1* transcripts in individual TCGA cancer tissues; these changes also vary between different cancer types. Most TCGA cancer tissues in 11 out of 15 cancer types mainly expressed the E27-included splice variants of *PBRM1*. In contrast, KIRC revealed a distributed AS pattern of *PBRM1* E27 in cancer tissues, while the adjacent normal tissues exhibited a significantly high E27 inclusion rate in the *PBRM1* mRNA. Further studies should aim at investigating the AS pattern alteration drivers of *PBRM1* E27 in cancer tissues as well as the unique splicing landscapes of ccRCC compared to other cancer types. Moreover, investigating the clinical impact of AS alterations of *PBRM1* E27, specifically targeting each cancer type, may provide a comprehensive understanding of cancer type-specific roles of PBRM1, providing a new biomarker to predict the therapeutic efficiency of immune checkpoint blockade.

## Methods

**Reagents and tools table**

| Antibodies | | | |
| --- | --- | --- | --- |
| **Name** | **Source** | **Catalog Number** | **Application** |
| PBRM1 | Cell Signaling Technology | 91894 | WB, 1:1000 IF, 1:500 IP, 2 μg/ sample ChIP, 2 μg/ sample |
| RBFOX2 | Bethyl Laboratories | A300-864A | WB, 1:4000 RNA IP, 1 μg/ sample |
| FLAG | Sigma-Aldrich | F3165 | WB, 1:20000 |
| Myc | Invitrogen | 46-0603 | WB, 1:5000 |
| PD-L1 | Cell Signaling Technology | 13684 | WB, 1:200 |

| Antibodies | | | |
|---|---|---|---|
| Name | Source | Catalog Number | Application |
| ARID2 | Santa Cruz Biotechnology | sc-166117 | WB, 1:200 IP, 2 µg/sample ChIP, 2 µg/sample |
| ARID1A | Cell Signaling Technology | 12354 | WB, 1:1000 |
| BRM | Cell Signaling Technology | 11966 | WB, 1:1000 |
| BRG1 | Cell Signaling Technology | 3508 | WB, 1:1000 |
| BAF170 | Cell Signaling Technology | 12760 | WB, 1:1000 |
| BAF155 | Cell Signaling Technology | 11956 | WB, 1:1000 |
| BAF47 | Cell Signaling Technology | 91735 | WB, 1:1000 |
| Tubulin | Sigma-Aldrich | T5168 | WB, 1:5000 |
| GAPDH | Meridian Bioscience | H86045P | WB, 1:2000 |

| Plasmid constructs | |
|---|---|
| Name | Backbone |
| pCMV6-Entry-RBFOX2 | pCMV6-Entry |
| pCS3-MT-RBFOX2 | pCS3 + MT |
| pEGFP-C3-Minigene-WT | pEGFP-C3 |
| pEGFP-C3-Minigene-Mut 1 | pEGFP-C3 |
| pEGFP-C3-Minigene-Mut 2 | pEGFP-C3 |
| pEGFP-C3-Minigene-Mut 3 | pEGFP-C3 |
| pEGFP-C3-Minigene-Mut all | pEGFP-C3 |
| pEGFP-C3-Minigene-Mut 1 (V2) | pEGFP-C3 |
| pGL3-basic-*PD-L1* promoter | pGL3-basic |

| Oligonucleotides | | |
|---|---|---|
| Name | Sequences (5′ → 3′) | Modifications |
| PCR primers | Table EV1 | N/A |
| siPBRM1 | (RNA) GAGUUGUC GGAAUAAC CAAUU | N/A |
| siE27-PBRM1 | (RNA) UGUUAGCAU GGGCAGCAUGC | N/A |
| siRBFOX2 #1 | (RNA) GGGAUUCGG GUUCGUAACU | N/A |

| Antibodies | | | |
|---|---|---|---|
| Name | Source | Catalog Number | Application |
| siRBFOX2 #2 | (RNA) AAUGAAC GUGGCUCUA AGGGAUU | N/A | |
| Control SSO | UGCAUUCG CCCUCUUAAU GGGGA | 2′-OME, PS | |
| PBRM1 SSO | GCCUUGAAGG ACACAACAA AUUUC | 2′-OME, PS | |
| sgRNA-PBRM1 E27 | (Top) CACCGT GTGTCCTTCAA GGCATGAT (Bottom) AAACATCAT GCCTTGAAG GACACAC | N/A | |
| *PBRM1* WT intron RNA | (RNA) UAGCUCAU GUUGCAUGU UAAUGAUGCA UGUCUGAAA UUU | 5′-end Biotinylation | |
| *PBRM1* Mut1/2 intron RNA | (RNA) UAGCUCAU GUACCUA CUUAAUG AACCUACUC UGAAAUUU | 5′-end Biotinylation | |

| Chemicals, Enzymes and other reagents | | |
|---|---|---|
| Reagent | Source | Catalog Number |
| DMEM | WELGENE | LM001-05 |
| RPMI 1640 | Corning | 10-040-CV |
| α-MEM | Gibco | 12561-056 |
| FBS | Gibco | 12483-020 |
| Penicillin/streptomycin | Corning | 30-002-CI |
| β-mercaptoethanol | Gibco | 21985-023 |
| Inositol | Sigma-Aldrich | I7508 |
| Folic acid | Sigma-Aldrich | F8758 |
| IL-2 | PeproTech | 200-02 |
| EGF | PeproTech | AF-100-15 |
| IFNγ | PeproTech | 300-02 |
| PolyMag | Chemicell | 9003 |
| jetPEI | Polyplus | 101000053 |
| M-MLV reverse transcriptase | Promega | M170B |
| RNase inhibitor | Enzynomics | M007S |
| SYBR Green PCR Master Mix | GENETBIO | Q-9200 |
| Hot-start PCR master mix | Bioneer | K-2630 |
| Apa1 | Takara | 1005A |
| DAPI | Thermo Fischer Scientific | D1306 |

| Antibodies | | | |
|---|---|---|---|
| **Name** | **Source** | **Catalog Number** | **Application** |
| **Other** | | | |
| **Name** | **Source** | **Catalog Number** | |
| Hybrid-R™ RNA extraction kit | GeneAll | 305-101 | |
| Dual-Glo Luciferase Assay System | Promega | E1910 | |
| Endometrial cancer cDNA array | ORIGENE | EDRT102 | |
| Breast cancer cDNA array | ORIGENE | BCRT101 | |
| Expin Gel SV mini kit | GeneAll | 102-102 | |
| TOPO TA cloning kit | Enzynomics | EZ011S | |
| TruSeq Stranded mRNA LT Sample Prep Kit | Illumina | RS-122-9005DOC | |
| Dynabeads protein G | Invitrogen | 10004D | |
| Direct-zol RNA mini prep kit | Zymo Research | R2052 | |
| TnT® Quick Coupled Transcription/Translation System | Promega | L2080 | |
| Streptavidin-conjugated agarose beads | Invitrogen | 15492-050 | |
| Dual-Glo Luciferase Assay System | Promega | E1910 | |
| LDH release assay kit | Invitrogen | C20301 | |
| Cell Line Nucleofector™ Kit V | Lonza Bioscience | VCA-1003 | |

## Cell culture

All cell lines used in this study were obtained from the American Type Culture Collection (ATCC). HeLa, U-2 OS, HEK293, and B16-F10 cells were cultured in Dulbecco's modified Eagle's medium (DMEM; WELGENE, LM001-05) supplemented with 10% foetal bovine serum (FBS; Gibco, 12483-020) and 1% penicillin/streptomycin (Corning, 30-002-CI). AGS, K562, MCF7, and MDA-MB-231 cells were cultured in RPMI 1640 medium (Corning, 10-040-CV) supplemented with 10% FBS (Gibco, 12483-020) and 1% penicillin/streptomycin (Corning, 30-002-CI). NK-92 cells were grown in alpha-minimum essential medium ($\alpha$-MEM; Gibco, 12561-056), supplemented with 12.5% FBS (Gibco, 12483-020), 12.5% horse serum (Gibco, 26050-088), 100 $\mu$M $\beta$-mercaptoethanol (Gibco, 21985-023), 200 $\mu$M inositol (Sigma-Aldrich, I7508), 20 $\mu$M folic acid (Sigma-Aldrich, F8758), 100 U/ml IL-2 (PeproTech, 200-02), and 1% penicillin/streptomycin (Corning, 30-002-CI).

## siRNA and SSO

Non-targeting control siRNA (SN-1012), siPBRM1 (5′-GAG UUG UCG GAA UAA CCA AUU-3′), and siE27-PBRM1 (5′-UGU UAG CAU GGG CAG CAU GC-3′) were purchased from Bioneer. siRBFOX2 was purchased from Bioneer (#1; 5′-GGG AUU CGG GUU CGU AAC U-3′) and Dharmacon (#2; 5′-AAU GAA CGU GGC UCU AAG GGA UU-3′). siRNAs were transfected into HeLa and U-2 OS cells using the PolyMag reagent (Chemicell, 9003), according to the manufacturer's instructions. For RBFOX2 or PBRM1 knock-down, cells were seeded in 6-well plates ($3 \times 10^5$ cells/well) and transfected with 2 $\mu$g siRNA for 48 h. MDA-MB-231 cells were transfected with 2 $\mu$g of siControl or siE27-PBRM1 for 48 h, using the Cell Line Nucleofector Kit V (Lonza Bioscience, VCA-1003) according to the manufacturer's instructions. Subsequently, the MDA-MB-231 cells underwent a second transfection under the same condition, followed by a 48 h incubation.

All SSOs were synthesised as chemically modified RNA with a phosphorothioate backbone and 2′-$O$-methyl modification at the 2′-sugar position. PBRM1 SSO (5′-GCC UUG AAG GAC ACA ACA AAU UUC-3′) was designed to target the 3′ splice site of *PBRM1* E27. Control SSO (5′-UGC AUU CGC CCU CUU AAU GGG GA-3′) was synthesised as a non-targeting control. Cells were seeded in a 6-well plate ($3 \times 10^5$ cells/well) and were transfected with 1 $\mu$g (for HeLa cells) or 2 $\mu$g (for AGS, K562, and MCF7 cells) SSOs using PolyMag reagent, according to the manufacturer's instructions.

## Plasmids and minigene assay

The expression plasmid for C-terminal FLAG-tagged *RBFOX2* (NCBI reference sequence: NP_001026865.1) was cloned into the pCMV6-entry vector. To synthesise RBFOX2 (TnT), Myc-tagged *RBFOX2* was cloned into the pCS3 + MT vector. *PBRM1* minigene constructs were cloned into the pEGFP-C3 vector using PCR products containing PBRM1 E27 with flanking E25 and E28. The primer sequences used for minigene cloning are listed in Table EV1.

To perform the minigene assay, cells seeded in the 6-well plate ($3 \times 10^5$ cells/well) were transfected with 50 ng PBRM1 minigene constructs and 0.5 $\mu$g RBFOX2-FLAG expression vector (for HeLa cells) or 2 $\mu$g siRBFOX2 (for U-2 OS cells) for 48 h. Plasmid vectors were transfected using jetPEI reagent (Polyplus, 101000053) following the manufacturer's protocol.

## Establishment of $\Delta$E27 HeLa cells

The optimised sgRNA sequences targeting the 3′ splice site of *PBRM1* E27 were designed using CRISPOR tool (http://crispor.tefor.net/) as follows: 5′-CAC CGT GTG TCC TTC AAG GCA TGA T-3′ (top) and 5′-AAA CAT CAT GCC TTG AAG GAC ACA C-3′ (bottom). DNA oligonucleotides were annealed using a PCR system and cloned into the pX330 vector, followed by transfection into HeLa cells. Four days later, the cells were plated at 0.5 cells per well in 96-well plates to obtain single-cell clones. The clones were screened via RT-PCR and immunoblot analysis, and $\Delta$E27 HeLa cell lines with no change in PBRM1 protein levels were selected for further experiments.

## TCGA data analysis

We obtained mRNA expression data from TCGA patients using the UCSC Xena data portal (Goldman et al, 2020) (https://xenabrowser.net/datapages/). To identify key cancer driver genes, we downloaded information on the somatic mutation status of TCGA cancer tissues from the UCSC Xena data portal and extracted genes included as cancer driver genes in the list on the IntOgen platform (Martínez-Jiménez et al, 2020) (https://www.intogen.org). PSI values of the alternative exons were downloaded from the SpliceSeq database (Ryan et al, 2016) (http://projects.insilico.us.com/TCGASpliceSeq/). Detailed data used in this study are presented in Datasets EV1–11.

## RT-PCR and qRT-PCR

Total RNA was extracted using the Hybrid-R™ RNA extraction kit (GeneAll, 305-101). For cDNA synthesis, 1 µg total RNA was reverse-transcribed using random hexamer, M-MLV reverse transcriptase (Promega, M170B), and RNase inhibitor (Enzynomics, M007S). The endometrial cancer cDNA array, (ORIGENE, EDRT102) and the breast cancer cDNA array (ORIGENE, BCRT101) were used to validate the PBRM1 E27 AS patterns. qRT-PCR was performed using the SYBR Green PCR Master Mix (GENETBIO, Q-9200) on an AriaMx real-time system (Agilent Technologies). PCR was performed using a hot-start PCR master mix (Bioneer, K-2630), followed by agarose gel electrophoresis. The PCR products were sequenced by Sanger sequencing following TOPO TA cloning (Enzynomics, EZ011S) to confirm that the PCR products represent the *PBRM1* splice variants. To quantify the *PBRM1* splice variants, the contrast of the DNA band images was adjusted to that of the outside region of the DNA bands exhibiting the minimum signals, and the DNA signals were quantified using the ImageJ software. To calculate the exon exclusion ratio, PCR band intensity was normalised to the PCR product size. The primers used for PCR are listed in Table EV1.

To identify the proportion of E26 and E27 included *PBRM1* splice variants from the cancer cDNA arrays, PCR products were extracted from the agarose gel pieces using the Expin Gel SV mini kit (GeneAll, 102-102), then further PCR-amplified using *PBRM1* E25 and E28 targeting primer pairs. The PCR products were purified using the Expin Gel SV mini kit, mixed with Apa1 restriction enzyme (Takara, 1005A), and incubated for 3 h at 37 °C in a PCR machine, followed by agarose gel electrophoresis, and the calculation of the intensity ratios between the 309 bp (E26 + E27− splice variant)- and 150 bp (E26− E27+ splice variant)-size bands.

## RNA sequencing library preparation and analysis

Total RNA was extracted from WT and ΔE27 HeLa cells using the Hybrid-R™ RNA extraction kit. After the quality check of RNA using Bioanalyzer 2100 (Agilent Technologies), 1 µg of the total RNA samples was processed using TruSeq Stranded mRNA LT Sample Prep Kit (Illumina, RS-122-9005DOC), according to the TruSeq Stranded mRNA Sample Preparation Guide, Part # 15031047 Rev. E. Libraries were sequenced using the Illumina NovaSeq platform. Next, raw RNA-seq reads were pre-processed to discard low quality bases and adapter sequences using Trimmomatic (Bolger et al, 2014) (v.0.36). The remaining clean reads were aligned to the human genome (hg38) using HISAT2 (v.2.2.1) with default parameters (Kim et al, 2019). To quantify gene expression levels, reads uniquely aligned to Cufflinks (v.2.2.1) were input (Trapnell et al, 2010). CuffDiff (v.2.2.1) was used to test differential expression (Trapnell et al, 2013). We defined differentially expressed genes using the following criteria: FPKM > 5 and FDR < 0.01. The genome sequence and the annotation of human (hg38) were obtained from the UCSC genome browser (https://genome.ucsc.edu). GSEA (v4.3.3) was conducted using hallmark gene sets with RNA-seq data (Table EV2). STRING (v.12.0) was used to analyse the protein-protein network based on the co-expression score (Szklarczyk et al, 2023).

## Immunoblot analysis

Whole cells were lysed using RIPA buffer (Thermo Fisher Scientific, 89900) containing a protease inhibitor cocktail. Proteins were quantified using the BCA assay (iNtRON Biotechnology, 21071) and denatured by adding 2× Laemmli sample buffer (Bio-Rad, 1610737) supplemented with 5% β-mercaptoethanol (Sigma-Aldrich, T9281). For immunoblot analysis, proteins were separated using sodium dodecyl sulfate (SDS)-polyacrylamide gel electrophoresis and transferred to a nitrocellulose membrane (Merck, Millipore, HATF00010), which was blocked with 5% skim milk (Bio-Rad, 1706404) in phosphate-buffered saline (PBS) supplemented with 0.05% Tween-20 (PBST) for 1 h at 25 °C. The blocked membrane was incubated with the primary antibody, diluted in blocking buffer for 16 h at 4 °C. After washing with PBST, the membrane was incubated with a horseradish peroxidase-conjugated secondary antibody for 1 h at 25 °C. After further washing with PBST, the proteins were detected with Super-Signal substrate (Thermo Fisher Scientific, 34076) using a WSE-6200H LuminoGraph II (ATTO). The primary antibodies used in this study were as follows: anti-PBRM1 (Cell Signaling Technology, 91894; 1:1000), anti-RBFOX2 (Bethyl Laboratories, A300-864A; 1:4000), anti-FLAG (Sigma-Aldrich, F3165; 1:20,000), anti-Myc (Invitrogen, 46-0603; 1:5000), anti-PD-L1 (Cell Signaling Technology, 13684; 1:200), anti-ARID2 (Santa Cruz Biotechnology, sc-166117; 1:200), anti-ARID1A (Cell Signaling Technology, 12354; 1:1000), anti-BRM (Cell Signaling Technology, 11966; 1:1000), anti-BRG1 (Cell Signaling Technology, 3508; 1:1000), anti-BAF170 (Cell Signaling Technology, 12760; 1:1000), anti-BAF155 (Cell Signaling Technology, 11956; 1:1000), anti-BAF47 (Cell Signaling Technology, 91735; 1:1000), anti-tubulin (Sigma-Aldrich, T5168; 1:5000), and anti-GAPDH (Meridian Bioscience, H86045P; 1:2000).

## Immunofluorescence

HeLa cells were seeded in a 4-well Chamber Slide ($3 \times 10^4$ cells/well; Nunc, 154526), fixed with 4% paraformaldehyde for 10 min, and permeabilised with 0.5% Triton X-100 in PBS for 15 min. Cells were blocked with blocking buffer (5% goat serum, 1% bovine serum albumin, and 0.05% Tween 20 in PBS) and stained with the anti-PBRM1 antibody (Cell Signaling Technology, 91894; 1:500) diluted in blocking buffer. Goat anti-rabbit IgG antibody conjugated to Alexa Fluor 488 (Thermo Fisher Scientific, A11008; 1:500) was used as a secondary antibody. Nuclei were stained with 4′,6-diamidino-2-phenylindole (DAPI; Thermo Fischer Scientific, D1306; 1:1,000). Images were captured using an LSM 880 microscope equipped with an Airyscan (ZEISS).

## Co-IP

HeLa cells seeded in a 150 mm culture dish ($4 \times 10^6$ cells/well) were lysed using IP lysis buffer [25 mM Tris-HCl (pH 7.5), 150 mM NaCl, 1 mM EDTA, 1% NP-40, 5% glycerol, 1 mM PMSF, and 1 mM $Na_3VO_4$] supplemented with a protease inhibitor cocktail. For IP, cell lysates were incubated with 2 μg PBRM1 (Cell Signaling Technology, 91894) or ARID2 (Cell Signaling Technology, 12354) antibody for 2 h at 4 °C with rotation, followed by the addition of 50 μl Dynabeads protein G (Invitrogen, 10004D) and further rotation for 2 h at 4 °C. The Dynabeads protein G/antibody/protein complex was washed with IP lysis buffer. After washing six times, the immunoprecipitated proteins were obtained by adding 1× Laemmli sample buffer to the immune complex and then subjected to immunoblot analysis.

## RIP-PCR

HeLa cells seeded in a 150 mm culture dish ($4 \times 10^6$ cells/well) were lysed using RIP lysis buffer [25 mM Tris-HCl (pH 7.5), 150 mM KCl, 5 mM EDTA, 0.5 mM DTT, and 0.5% NP-40] containing protease inhibitor cocktail and RNase inhibitor. Cell lysates were incubated with 1 μg RBFOX2 antibody (Bethyl Laboratories; A300-864A) or control rabbit IgG (Sigma-Aldrich, I5006) for 4 h at 4 °C, with rotation, followed by further incubation with Dynabeads protein G (Invitrogen) for 2 h at 4 °C. The beads were washed six times with RIP buffer, after which total RNA was extracted using the Direct-zol RNA mini prep kit (Zymo Research, R2052) and reverse-transcribed with random hexamers and M-MLV reverse transcriptase. The primers used in RIP-PCR were as follows: 5′-CTA ATT CAT CCC AAA ATG TTG GGT C-3′ and 5′-CCA TCA TGC CTT GAA GGA CAC-3′.

## RNA pull-down assay

The RBFOX2 (TnT) protein was produced through in vitro cell-free protein expression using The TnT® Quick Coupled Transcription/Translation System (Promega, L2080), according to the manufacturer's instruction. The reaction was performed using 1 μg *RBFOX2* expression plasmid (Myc-tagged RBFOX in pCS3 + MT) in a 25 μl reaction volume for 90 min at 30 °C. Biotinylated *PBRM1* intron RNA probes (WT; 5′-[biotin]-UAG CUC AUG U<u>UG CAU GUU</u> AAU GA<u>U GCA UGU</u> CUG AAA UUU-3′, Mut1/2; 5′-[biotin]-UAG CUC AUG U<u>AC CUA C</u>UU AAU GA<u>A CCU ACU</u> CUG AAA UUU-3′) were ordered from Bioneer. Biotinylated *PBRM1* intron RNA probes (500 pmol) were incubated with RBFOX2 (TnT) protein (1.5 μl) in NP-40 buffer [50 mM Tris-HCl (pH 7.5), 150 mM NaCl, 5 mM EDTA, 0.1% NP-40, and protease inhibitor cocktail] for 3 h at 4 °C with rotation, followed by an additional incubation with streptavidin-conjugated agarose beads (Invitrogen, 15492-050) for 1 h at 4 °C. The pull-down complex was washed five times with NP-40 buffer, and the pulled-down RBFOX2 protein was analysed using immunoblot analysis.

## Dual-luciferase assay

The *PD-L1* promoter and the adjacent intronic region to exon 1 were amplified to generate a PCR product, which was then combined with another PCR product containing *PD-L1* exon 2 and 3′ intronic sequences, using overlap extension PCR. The resulting product was cloned into the pGL3-basic vector to construct the *PD-L1* promoter reporter vector. The primer sequences used for cloning are listed in Table EV1. Cells were seeded in 12-well plates ($1 \times 10^5$ cells/well) for 24 h, co-transfected with 470 ng *PD-L1* promoter reporter vector and 30 ng pRL Renilla vector for 24 h using jetPEI reagent following the manufacturer's protocol. Subsequently, the cells were treated with 20 ng/ml EGF (Pepro-Tech, AF-100-15) and 100 ng/ml IFNγ (PeproTech, 300-02) for 12 h and then subjected to a dual-luciferase assay using the Dual-Glo Luciferase Assay System (Promega, E1910).

## ChIP

HeLa cells seeded in a 150 mm culture dish ($4 \times 10^6$ cells/well) were fixed with 0.5% formaldehyde for 5 min at 37 °C and harvested and lysed in SDS lysis buffer [50 mM Tris-HCl (pH 8.0), 10 mM EDTA, and 1% SDS] supplemented with the protease inhibitor cocktail. Chromatin was sheared by sonication to yield DNA fragments of 300–600 bp. Sonicated chromatin solution was diluted 5-fold with ChIP dilution buffer [16.7 mM Tris-HCl (pH 8.0), 167 mM NaCl, 1.2 mM EDTA, 1.1% Triton X-100, and 0.01% SDS] and centrifuged at $16{,}000 \times g$ for 10 min at 4 °C. The supernatant was incubated with 2 μg anti-PBRM1 (Cell Signaling Technology; 91894) or anti-ARID2 (Cell Signaling Technology; 12354) antibody for 3 h at 4 °C with rotation, followed by the addition of 50 μl Dynabeads protein G and further rotation for 2 h at 4 °C. The immune complex was sequentially washed with low-salt wash buffer [20 mM Tris-HCl (pH 8.0), 150 mM NaCl, 2 mM EDTA, 1% Triton X-100, and 0.1% SDS], high-salt wash buffer [20 mM Tris-HCl (pH 8.0), 500 mM NaCl, 2 mM EDTA, 1% Triton X-100, and 0.1% SDS], LiCl wash buffer [10 mM Tris-HCl (pH 8.0), 0.25 M LiCl, 1 mM EDTA, 1% IGEPAL-CA630, and 1% deoxycholic acid], and TE buffer [10 mM Tris-HCl (pH 8.0) and 1 mM EDTA] twice. The beads were resuspended in elution buffer (1% SDS and 0.1 M $NaHCO_3$), and the supernatant was reverse-crosslinked overnight at 65 °C after adding 5 M NaCl (final concentration of 200 mM) with RNase A. Proteinase K was added to the samples and incubated at 60 °C for 1 h. Chromatin DNA was purified using a Expin Gel SV mini kit and used as a qRT-PCR template to amplify the *PD-L1* promoter region. The primers used in ChIP-qPCR were as follows: *PD-L1* promoter region, 5′-GGT AGG GAG CGT TGT TCC TC-3′ and 5′-CTC CTC TCT CCA TCC CAA AG-3′; *PD-L1* negative control region, 5′-CTG GCT ATA GTA TCC TAC TCT GG-3′ and 5′-TTA GGA GCT GTT TCT GTA GAG C-3′.

## NK-mediated cancer cytotoxicity assay

Target cancer cells (5000 cells) were trypsinised and co-cultured with NK-92 cells in 96-well round-bottom cell-floater plates (SPL, 34896) at 37 °C for 7 h. Next, cells were resuspended by pipetting and centrifuged at $300 \times g$ for 3 min. The supernatant was subjected to a lactate dehydrogenase (LDH) release assay (Invitrogen, C20301) to analyse the cancer-specific cytotoxicity. Cytotoxicity was calculated using the following formula: % specific lysis = (Sample LDH activity – spontaneous LDH activity)/(Maximal LDH activity – spontaneous LDH activity) × 100.

## Mouse tumour model

Mice were obtained from Daehan Biolink and maintained under a 12 h light/dark cycle at $22 \pm 2\,°C$ and 40–60% humidity. Trypsinised B16-F10 cells ($3 \times 10^6$ cells) were transfected with 4 µg of control SSO or PBRM1 SSO using Cell Line Nucleofector™ Kit V (Lonza Bioscience, VCA-1003) and Nucleofector™ I device, according to the manufacturer's instructions. The cells were seeded in a 150 mm culture dish and incubated for 2 days. Transfected B16-F10 cells were injected subcutaneously ($5 \times 10^5$ cells/100 µl PBS per mouse) into the backs of 6-week-old male C57BL/6 mice, and the remaining cells were used to confirm the *PBRM1* E27 AS pattern. Nine days later, the mice were euthanised using carbon dioxide inhalation, and tumour sizes were measured. Tumour volume was calculated using the following formula: Volume = $1/6\pi \times$ (length) $\times$ (width)$^2$. The animal experiment was approved by the Institutional Animal Care and Use Committee of Chungnam National University (202006A-CNU-089).

## Statistical analysis

Box plots represent the median, quartiles, 10th percentile, and 90th percentile. Bar graphs are presented as mean ± SD. qRT-PCR was conducted using biological replicates derived from independent samples, and mRNA expression levels were calculated as fold changes relative to the negative controls. Statistical analyses, including one-way analysis of variance (ANOVA) and Log-rank (Mantal-Cox) tests for survival analysis, were performed using GraphPad Prism software. Two-tailed Student's *t*-test and Pearson correlation coefficients with *p*-values were calculated using Excel.

# Data availability

The data generated in this study are available in the Article and its Supplementary Data. All raw RNA sequencing data have been deposited in the NCBI SRA database with the accession numbers SRX20520582, SRX20520583, SRX20520584, and SRX20520585 under BioProject PRJNA976477 (https://www.ncbi.nlm.nih.gov/bioproject/?term=PRJNA976477).

The source data of this paper are collected in the following database record: biostudies:S-SCDT-10_1038-S44318-024-00262-7.

# Peer review information

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

## Acknowledgements

This work was supported by the National Research Foundation of Korea (NRF-2022R1A2C1003870 and RS-2024-00409098).

## Author contributions

**Namjoon Cho**: Conceptualization; Investigation; Visualization; Writing—original draft; Writing—review and editing. **Seung-Yeon Kim**: Resources; Investigation. **Sung-Gwon Lee**: Formal analysis; Writing—original draft. **Chungoo Park**: Conceptualization; Investigation. **Sunkyung Choi**: Validation; Investigation; Writing—review and editing. **Eun-Mi Kim**: Conceptualization; Resources; Methodology; Writing—review and editing. **Kee K Kim**: Conceptualization; Supervision; Funding acquisition; Methodology; Writing—original draft; Project administration; Writing—review and editing.

Source data underlying figure panels in this paper may have individual authorship assigned. Where available, figure panel/source data authorship is listed in the following database record: biostudies:S-SCDT-10_1038-S44318-024-00262-7.

## Disclosure and competing interests statement

The authors declare no competing interests.

# Expanded View Figures

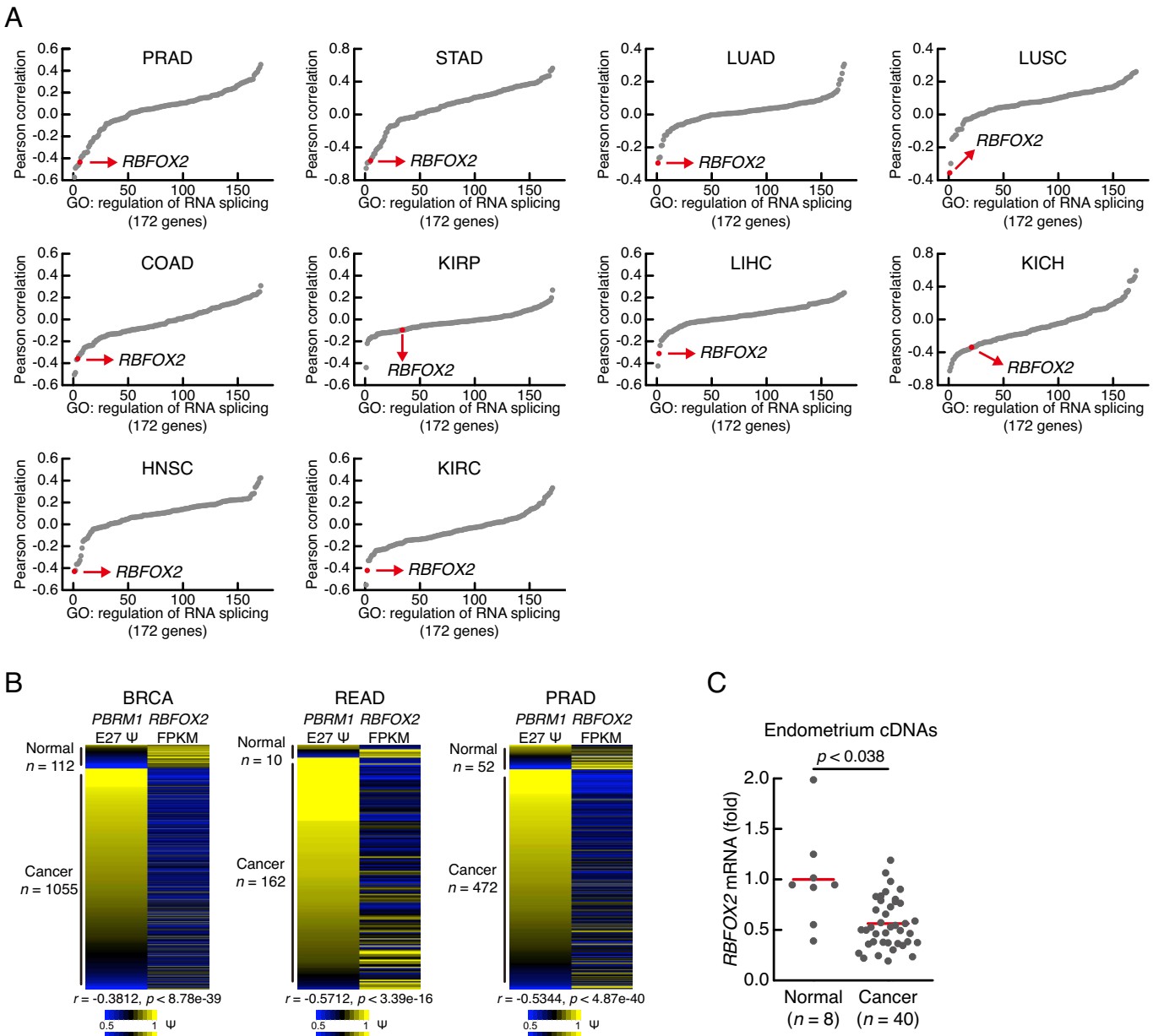

**Figure EV1.** ***RBFOX2* mRNA levels inversely correlate with PSI values of *PBRM1* E27 in cancer tissues.**

(**A**) Dot plots of splicing factors showing Pearson's correlation coefficients between FPKM values of the splicing factors and PSI values of *PBRM1* E27 in individual TCGA cancer tissues. *RBFOX2* is indicated with a red dot. (**B**) Heatmaps showing the inverse correlation between the PSI values of *PBRM1* E27 and mRNA levels of *RBFOX2* in BRCA, READ, and PRAD. Pearson's correlation coefficients and *p*-values are shown at the bottom of the heatmap. *n*, number of samples; Ψ, PSI. (**C**) *RBFOX2* mRNA levels in endometrium cDNAs. The red lines represent the means. *n*, number of samples. *p*-values were calculated using a two-tailed Student's *t*-test. Source data are available online for this figure.

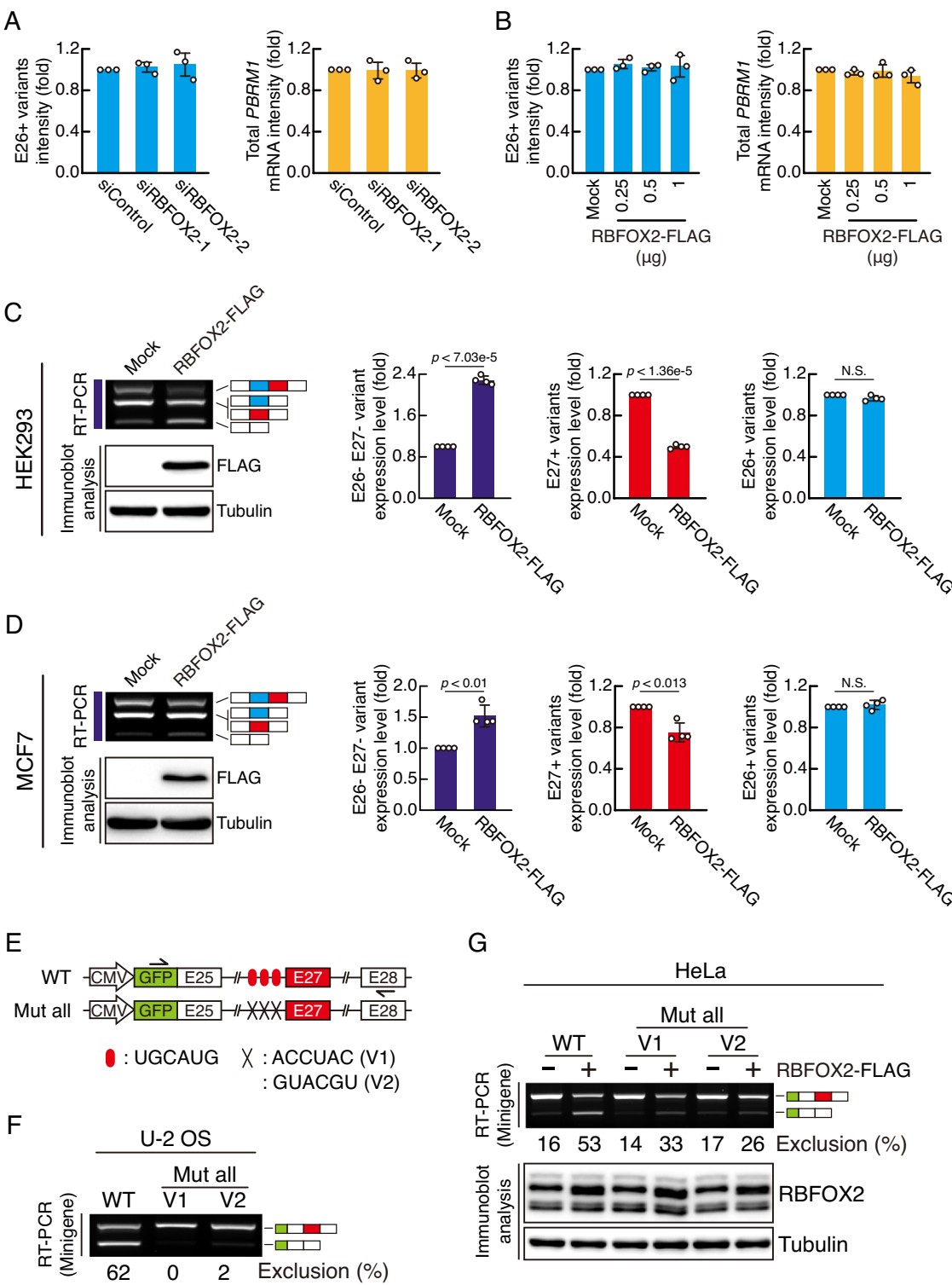

**Figure EV2. RBFOX2 represses E27 inclusion in *PBRM1* mRNA.**

(A, B) Quantification of RT-PCR products shown in Fig. 2F, G for E26+ variants (left) and total *PBRM1* mRNA levels in *RBFOX2*-knockdown U-2 OS cells (A) or in RBFOX2-FLAG-overexpressed HeLa cells (B) (*n* = 3, biological replicates). Bars indicate mean ± SD. (C, D) AS patterns of *PBRM1* in RBFOX2 overexpressed cells. HEK293 (C) and MCF7 (D) cells were transfected with RBFOX2 expression vectors for 36 h, followed by RT-PCR (top left), immunoblot (bottom left), and qRT-PCR (right) analyses (*n* = 4, biological replicates). Bars indicate mean ± SD. *p*-values were calculated using a two-tailed Student's *t*-test. N.S., not significant. (E) Schematic representation of *PBRM1* minigene constructs. (F) AS pattern of *PBRM1* minigene constructs in U-2 OS cells. (G) RT-PCR (top) and immunoblot (bottom) analyses for the AS pattern of *PBRM1* minigene transcripts by RBFOX2-FLAG overexpression in HeLa cells. Source data are available online for this figure.

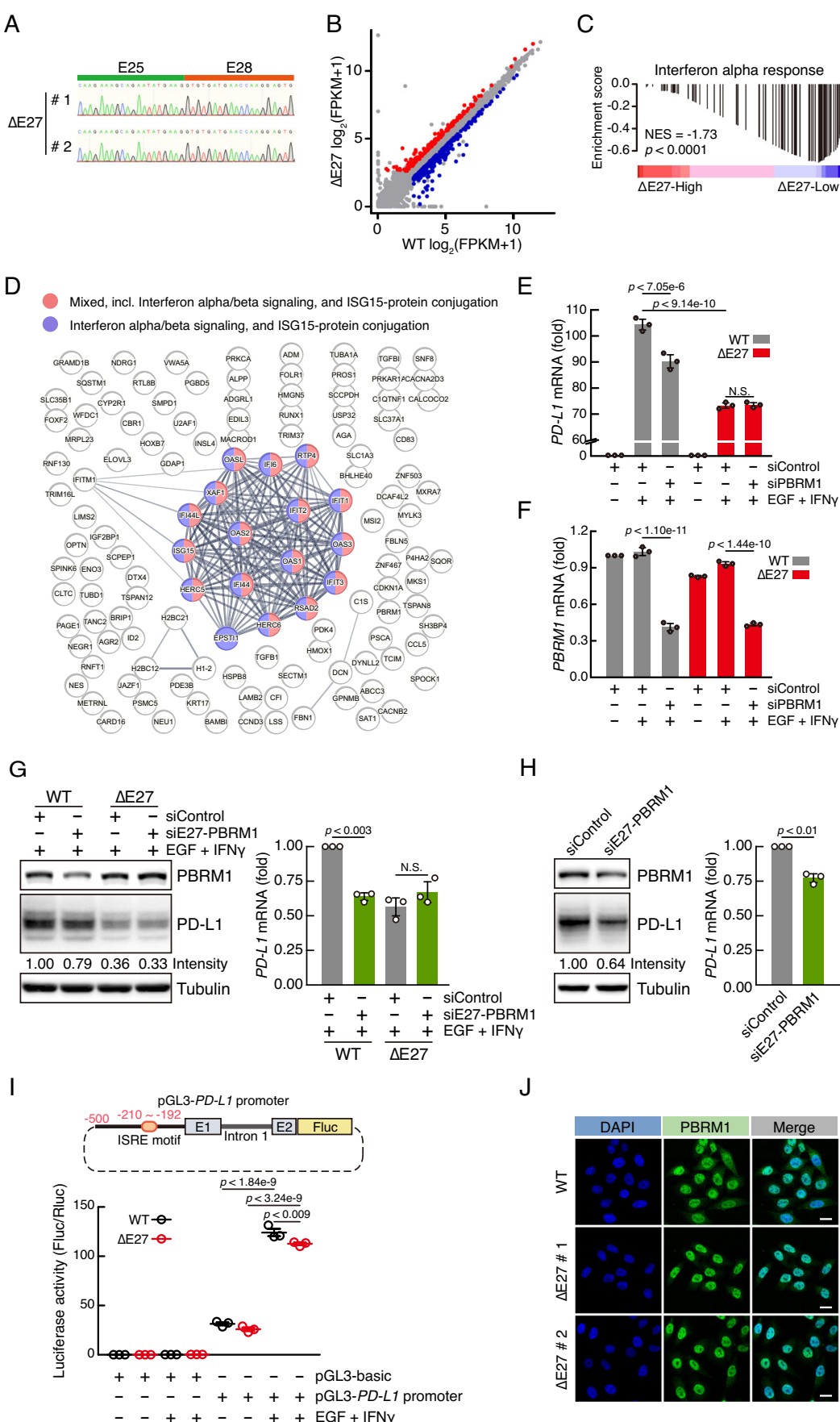

◀ **Figure EV3. PBRM1 differentially affects gene expression based on E27 AS.**

(A) Sanger sequencing with RT-PCR products of skipping variant, lacking E26 and E27, of *PBRM1* expressed in ΔE27 HeLa cell lines. (B) Scatter plot of RNA sequencing data displaying upregulated (red) and downregulated (blue) genes in ΔE27 HeLa cells compared to those in WT HeLa cells. (C) GSEA plot of IFNα-responsive genes in all gene transcripts of ΔE27 HeLa cells versus WT HeLa cells. The nominal *p*-value is presented. (D) STRING protein-protein network analysis of the top 120 downregulated genes in ΔE27 HeLa cells. The lines indicate the co-expression network between proteins and the line thickness indicates the strength of data support. (E, F) qRT-PCR analysis of *PD-L1* (E) or *PBRM1* (F) mRNA levels. HeLa cells were treated with 20 ng/ml EGF and 100 ng/ml IFNγ for 12 h following transfection with siPBRM1 for 36 h ($n = 3$, biological replicates). Bars indicate mean ± SD. *p*-values were calculated using one-way ANOVA with Dunnett's multiple comparison test. (G, H) Immunoblot (left) and qRT-PCR (right) analyses confirming PD-L1 expression levels following knockdown of the E27-included PBRM1 isoforms using siRNA targeting PBRM1 E27 sequences (siE27-PBRM1) in HeLa (G) or MDA-MB-231 cells (H). $n = 3$, biological replicates. Bars indicate mean ± SD. *p*-values were calculated using two-tailed Student's *t*-test. N.S., not significant. (I) *PD-L1* promoter reporter assay in WT and ΔE27 HeLa cells. The schematic shows the *PD-L1* promoter reporter vector (top), and the graph presents the results of firefly luciferase (Fluc) activities normalised to renilla luciferase (Rluc) activities (bottom). $n = 3$, biological replicates. Lines indicate the mean. *p*-values were calculated using one-way ANOVA with Bonferroni's multiple comparison test. ISRE, Interferon-sensitive response element. (J) Immunofluorescence analysis of PBRM1 (green) and DAPI (nuclei; blue) in WT and ΔE27 HeLa cells. Scale bars, 20 μm. Source data are available online for this figure.

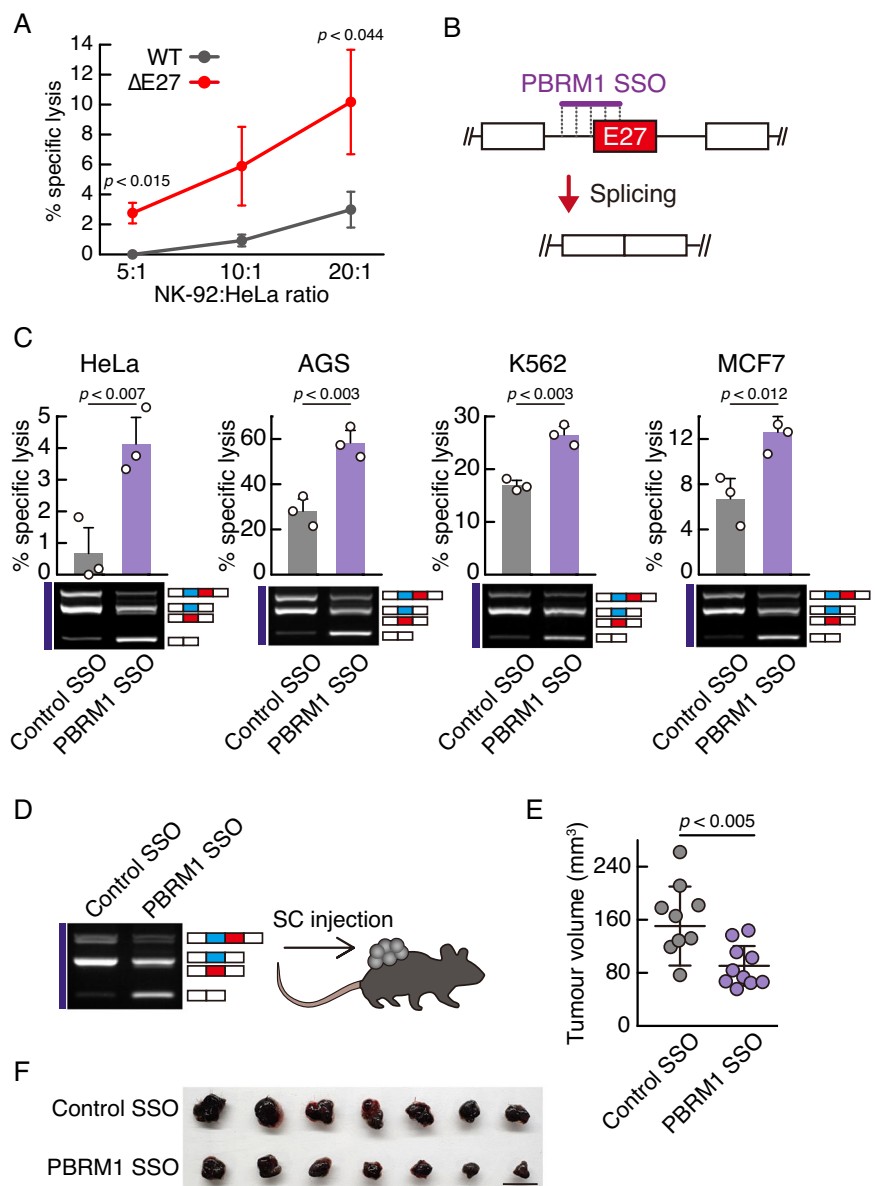

**Figure EV4. SSO-induced E27 exclusion of *PBRM1* suppresses cancer immune evasion.**

(**A**) NK-92 cell cytotoxicity against ΔE27 HeLa cells. HeLa cells were co-cultured with NK-92 cells for 7 h ($n = 3$, biological replicates). *p*-values were calculated using one-tailed Student's *t*-test. (**B**) Design of PBRM1 SSO for induction of E27 exclusion in *PBRM1*. (**C**) NK-92 cell cytotoxicity against PBRM1 SSO-transfected cancer cells (top). NK-92 cells were co-cultured with cancer cells at an effector:target ratio of 20:1 for 7 h ($n = 3$, biological replicates). SSO-induced *PBRM1* E27 exclusion was confirmed using RT-PCR (bottom). Bars indicate mean ± SD. *p*-values were calculated using one-tailed Student's *t*-test. (**D**) RT-PCR analysis confirmed *PBRM1* E27 exclusion by PBRM1 SSO transfection into B16-F10 cells used for subcutaneous injection into mice. (**E, F**) Tumour volume (**E**) and representative images of tumour (**F**) from C57BL/6 mice injected subcutaneously with control SSO- ($n = 9$, individual mice) or PBRM1 SSO-transfected B16-F10 cells ($n = 10$, individual mice) on day 9. The *p*-value was calculated using two-tailed Student's *t*-test. Scale bar, 1 cm. Source data are available online for this figure.

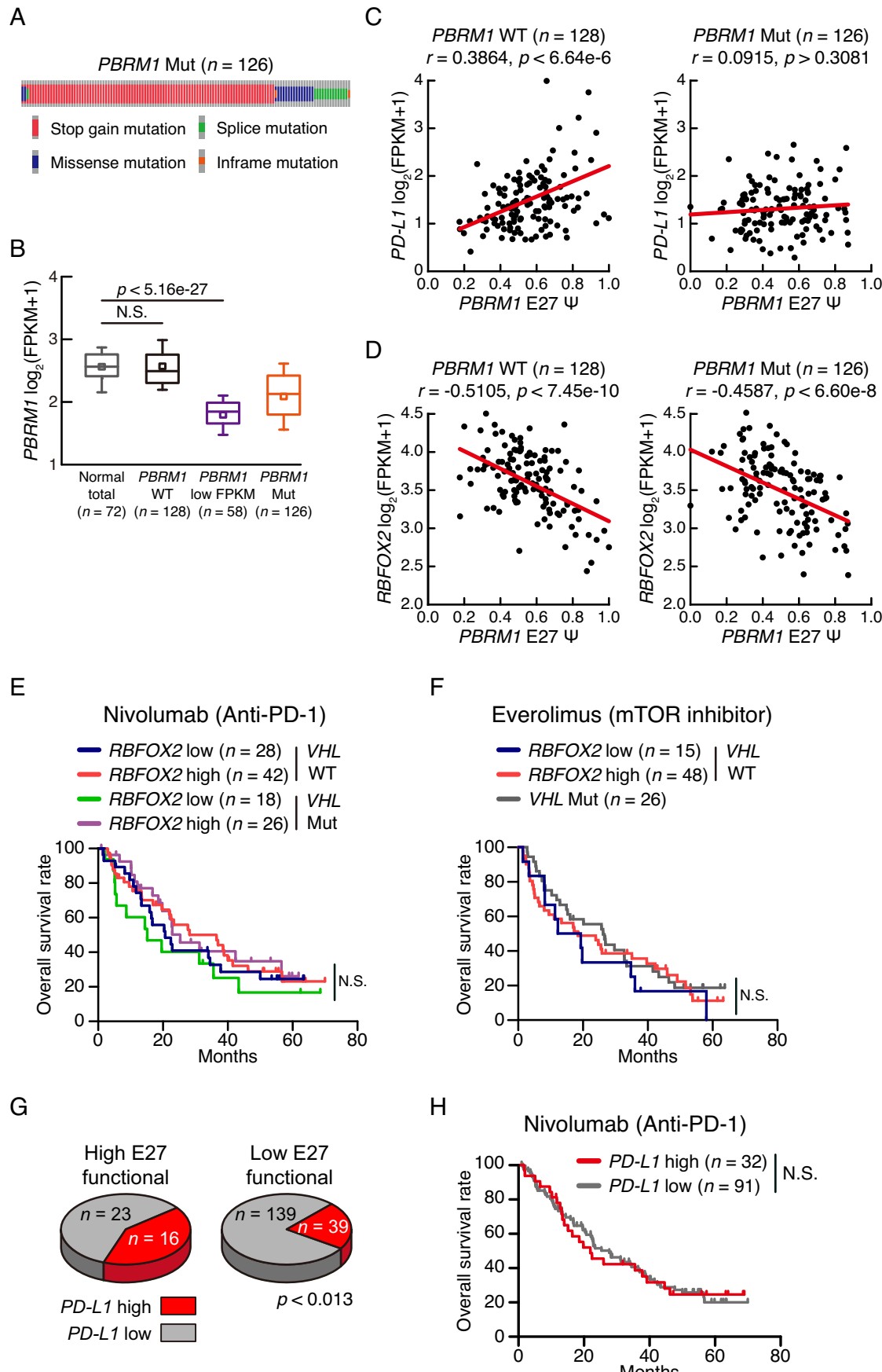

◀

**Figure EV5. E27 inclusion rates in *PBRM1* correlate with *PD-L1* expression in *PBRM1* WT ccRCC but not in *PBRM1* Mut ccRCC.**

(A) Mutation frequency in the *PBRM1* gene across patients with KIRC in the *PBRM1* Mut group. Boxes indicate cancer tissues, with colours representing the four somatic mutation statuses. (B) Box plot showing *PBRM1* mRNA level in normal and ccRCC tissues. Boxes represent the median, quartiles, 10th percentile, and 90th percentile. *n*, number of samples. *p*-values were calculated by the one-way ANOVA with Bonferroni's multiple comparison test. (C, D) Scatter plot of *PBRM1* E27 PSI versus *PD-L1* mRNA level (C) or *RBFOX2* mRNA level (D) in individual *PBRM1* WT ccRCC (left) or *PBRM1* Mut ccRCC (right) cancer tissues. Pearson correlation coefficients and *p*-values are presented. *n*, number of samples; Ψ, PSI. (E, F) Kaplan–Meier survival curves of patients with ccRCC treated with nivolumab (E) or everolimus (F). Patients with ccRCC possessing the non-mutant *VHL* gene and a low *VHL* mRNA level (below 12%) were not classified in the *VHL* WT group. *n*, number of samples. (G) Pie charts showing the ratios of ccRCC patients with high and low *PD-L1* levels in high (*RBFOX2* low and *PBRM1* WT) or low (all others) E27 functional groups. Patients with ccRCC treated with nivolumab or everolimus were classified. *n*, number of samples. The *p*-value was calculated using chi-squared test. (H) Kaplan–Meier survival curve of ccRCC patients treated with nivolumab, stratified by *PD-L1* expression status. *n*, number of samples. N.S., not significant.

