## [Peer Review File · The EMBO Journal]

Alternative splicing of PBRM1 mediates resistance to PD-1 blockade therapy in renal cancer

Namjoon Cho, Seung-Yeon Kim, Sung-Gwon Lee, Chungoo Park, Sunkyung Choi, Eun-Mi Kim, and Kee K. Kim

Corresponding author(s): Kee K. Kim (kimkk@cnu.ac.kr) , Eun-Mi Kim (eunmi.kim@swu.ac.kr), Sunkyung Choi (skchoi@kmu.ac.kr)

Review Timeline:

Submission Date:	8th Feb 24
Editorial Decision:	7th Apr 24
Revision Received:	4th Jul 24
Editorial Decision:	21st Aug 24
Revision Received:	4th Sep 24
Accepted:	20th Sep 24

Editor: Daniel Klimmeck

Transaction Report:

Dear Dr Kim,

Thank you again for the submission of your manuscript (EMBOJ-2024-116918) to The EMBO Journal. As mentioned earlier, your study was assessed by three reviewers with expertise in cancer immunology and splicing, whose comments are enclosed below.

As you will see from the experts' reports, the referees acknowledge the analysis and potential interest and value of your findings. However, they also express major concerns regarding completeness and endogenous relevance of the findings, which need to be addressed thoroughly to make them supportive of publication in the EMBO Journal. The reviewers also raise a number of issues related to the presentation of the findings, additional controls and improved methods annotation required, statistics applied, that would need to be conclusively addressed to achieve the level of robustness and clarity needed for The EMBO Journal.

Given the overall interest stated and broader angle of your findings, we are able to invite you to revise your manuscript experimentally to address the referees' comments. I need to stress though that we do require strong support from the referees on a revised version of the study in order to move on to publication of the work.

In light of the extensive experimentation requested, I would appreciate if you could contact me during the next weeks for exchange e.g. a video call to discuss your perspective on the comments and potential plan for revisions.

Please feel free to contact me if you have any questions or need further input on the referee comments.

When submitting your revised manuscript, please carefully review the instructions below.

Please feel free to approach me any time should you have additional questions related to this.

Thank you for the opportunity to consider your work for publication.

I look forward to your revision.

Kind regards,

Daniel Klimmeck

Daniel Klimmeck, PhD
Senior Editor
The EMBO Journal

Instruction for the preparation of your revised manuscript:

- 1) a .docx formatted version of the manuscript text (including legends for main figures, EV figures and tables). Please make sure that the changes are highlighted to be clearly visible.
- 2) individual production quality figure files as .eps, .tif, .jpg (one file per figure).
- 3) a .docx formatted letter INCLUDING the reviewers' reports and your detailed point-by-point response to their comments. As part of the EMBO Press transparent editorial process, the point-by-point response is part of the Review Process File (RPF), which will be published alongside your paper.
- 4) a complete author checklist, which you can download from our author guidelines ([https://wol-prod-cdn.literatumonline.com/pb-assets/embo-site/Author Checklist%20-%20EMBO%20J-1561436015657.xlsx](https://wol-prod-cdn.literatumonline.com/pb-assets/embo-site/Author%20Checklist%20-%20EMBO%20J-1561436015657.xlsx)). Please insert information in the checklist that is also reflected in the manuscript. The completed author checklist will also be part of the RPF.

6) It is mandatory to include a 'Data Availability' section after the Materials and Methods. Before submitting your revision, primary datasets produced in this study need to be deposited in an appropriate public database, and the accession numbers and database listed under 'Data Availability'. Please remember to provide a reviewer password if the datasets are not yet public (see <https://www.embopress.org/page/journal/14602075/authorguide#datadeposition>).

7) Our journal encourages inclusion of *data citations in the reference list* to directly cite datasets that were re-used and obtained from public databases. Data citations in the article text are distinct from normal bibliographical citations and should directly link to the database records from which the data can be accessed. In the main text, data citations are formatted as follows: "Data ref: Smith et al, 2001" or "Data ref: NCBI Sequence Read Archive PRJNA342805, 2017". In the Reference list, data citations must be labeled with "[DATASET]". A data reference must provide the database name, accession number/identifiers and a resolvable link to the landing page from which the data can be accessed at the end of the reference. Further instructions are available at .

8) At EMBO Press we ask authors to provide source data for the main and EV figures. Our source data coordinator will contact you to discuss which figure panels we would need source data for and will also provide you with helpful tips on how to upload and organize the files.

Numerical data can be provided as individual .xls or .csv files (including a tab describing the data). For 'blots' or microscopy, uncropped images should be submitted (using a zip archive or a single pdf per main figure if multiple images need to be supplied for one panel). Additional information on source data and instruction on how to label the files are available at .

9) We replaced Supplementary Information with Expanded View (EV) Figures and Tables that are collapsible/expandable online (see examples in <https://www.embopress.org/doi/10.15252/embj.201695874>). A maximum of 5 EV Figures can be typeset. EV Figures should be cited as 'Figure EV1, Figure EV2" etc. in the text and their respective legends should be included in the main text after the legends of regular figures.

11) For data quantification: please specify the name of the statistical test used to generate error bars and P values, the number (n) of independent experiments (specify technical or biological replicates) underlying each data point and the test used to calculate p-values in each figure legend. The figure legends should contain a basic description of n, P and the test applied. Graphs must include a description of the bars and the error bars (s.d., s.e.m.).

We realize that it is difficult to revise to a specific deadline. In the interest of protecting the conceptual advance provided by the work, we recommend a revision within 3 months (6th Jul 2024). Please discuss the revision progress ahead of this time with the editor if you require more time to complete the revisions. Use the link below to submit your revision:

Referee #1:

In this manuscript, Cho et al. investigate alternative splicing (AS) alterations within cancer driver genes. Their analysis uncovers AS alterations of PBRM1 exon 27 across various types of cancer tissues. Additionally, they present a regulatory mechanism of AS in PBRM1 and its potential implications for gene expression and clinical outcomes in cancer patients.

This manuscript is well-written and addresses an important topic, given the limited knowledge about functional differences among splicing isoforms in cancer tissues. Notably, this study highlights that differences in AS, even in the absence of differences in protein expression, can impact both gene expression and clinical outcomes. Consequently, this manuscript provides a new perspective on cancer by exploring AS alterations in depth.

Major concerns

- Figure 1: qPCR should be used to assess the AS pattern of PBRM1 E27 usage, normalized to gene expression, in human tissue samples.
- Additionally, the current approach appears insufficient to distinguish between E26 or E27 exclusion. Accordingly, figure 1E requires adjustment.
- Figure 2: It is unclear how the current design of primers can distinguish transcripts with or without E26 or E27. Accordingly, the primer design should be clarified as primers should cover exon junctions to be specific.
- The methods for quantification should be described. The changes observed on the gel do not appear as drastic as the quantification suggests.

Minor concerns

- The terminology used in Figure 2F and 2G is confusing. What exactly does "skipping isoform" refer to? Similarly, what does "E27+ isoform" signify?
- Are transcripts with differential E27 exon usage associated with specific splice isoforms? RSEM analysis can be utilized to estimate isoform expression levels from RNA-seq data.
- eCLIP studies have been conducted, reporting RBFOX2 target genes. Is PBRM1 a known RBFOX2 target gene?
- RBFOX2 is an important regulator of AS in human embryonic stem cells, which should be addressed in the discussion section.
- Interestingly, PBRM1 E27, was significantly more included in cancer tissues than in normal tissues, whereas kidney renal cell carcinoma showed E27 exclusion. This finding should be addressed in the discussion section.

Referee #2:

Cho et al. describe a novel PBRM1 splicing isoform that impacts PD-L1 expression in tumors. This is a highly exciting and well-crafted manuscript. The authors go full circle, starting with the identification of splicing variants of cancer-relevant genes using genomic data. From there, they selected PBRM1 E27 for further analysis. They confirmed the alterations in cancer cells, determined that RBFOX2 regulates PBRM1 E27 splicing, and evaluated the downstream effects and clinical relevance. The findings are important since they provide a path to improve immunotherapy success. Regulation of PBRM1 splicing via AS oligos causes subsequent alterations in PD-L1 levels. I have just a few suggestions.

- Are there any public CLIP datasets that show RBFOX2 binding to PBRM1?
- A table with the results of the RNA seq analysis should be included in Sup data.
- The analysis shown in Fig 4b is very superficial. A more detailed GO, pathway, and network analysis should be performed, and a table with complete results should be included in the supplementary materials.

Referee #3:

The authors present a nice study on the regulation of PBRM1 alternative splicing in cancer cells. Splicing factor RBFOX2 directly binds to PBRM1 intron, promoting Exon 27 skipping. Exclusion of Exon 27 abolishes the binding of PBRM1 to PD-L1 promoters, resulting in reduced PD-L1 expression and enhanced response to immune therapy. Cancer cells exhibit significant alterations in splicing, generating potentially novel proteins for their benefit. Manipulating splicing has emerged as a promising therapeutic strategy for cancer treatment. This study exemplifies how immune checkpoint blockade therapy can be improved by modulating altered splicing in cancer cells. The analysis of splicing events appears to be well-performed and robust. Here are some suggestions could potentially improve the study:

Major points:

1. The authors should elaborate more on the alternative splicing analysis of E26 and E27 in Figure 1, as the current description is unclear. Are E26 and E27 mutually exclusive or independent cassette exons? Additionally, the authors could design new pairs of primers, e.g., spanning E25/E26 or E25/27, to confirm rigorously that the upper band in Figure 1e represents the E27-containing isoform, similar to subsequent figures. Alternatively, the PCR product can be topo-cloned and Sanger sequenced for validation. While the PCR strategy in Figure 2 is sound, it is limited to two selected cell lines.
2. In Figure 3, while minigene reporters are valuable tools for studying alternative splicing, they may not fully reflect endogenous regulatory mechanisms. It is suggested to use CRISPR to modify/mutate the RBFOX2 binding motifs and probe splicing changes, which would strengthen the mechanistic study. The authors should clarify how they determined the mutational sequences as ACCTAC and explore if other mutations yield similar effects. Are there any RBFOX2 binding motifs downstream of E27 that might also influence E27 inclusion?
3. Can the authors validate the mechanisms in multiple cancer cell lines:
 - Overexpression of RBFOX2 in RBFOX2-low cell lines promotes PBRM1 Exon 27 skipping;
 - knockdown of E27-containing isoforms in tumor cells reduces PD-L1 expression.
4. In Figure 5D, the enrichment of PD-L1 binding does not significantly differ between negative control regions and PD-L1-negative control regions (%input), raising concerns about the specificity of this binding. Are there previous datasets available to corroborate the binding of PBRM1 to the PD-L1 promoter? Is PBRM1 a direct DNA-binding factor with known DNA motifs, and if so, do they exist in the PD-L1 promoter? In addition to qPCR, the authors should present gel plots of CHIP-amplified regions, showing a single specific band with differences in intensity as supporting evidence, similar to what they did in Figure 3b RIP experiment. Luciferase reporter assays with the PD-L1 promoter could also demonstrate the functions of potential binding.

Minor points:

1. In Figure 4d, why are there no error bars in the RNA-seq data?
2. How many PBRM1 isoforms exist in terms of Exon 26 and 27 inclusion/skipping? Does the $\Delta 27$ form in the engineered HeLa cell line contain E26 or not?
3. What are the PD-L1 expression levels and responses to PD-1 blockade therapy in KIRC? Does reduced inclusion of E27 correlate with a better response to immune therapy?
4. For most qPCR quantifications in the paper, the data points in all control bars are consistently the same without variation. The authors should explain how the quantification was performed.
5. In the analysis of KIRC TCGA data, how many mutation types are included in the PBRM1 Mut group? Are all of them loss-of-function mutations?"

Point-by-Point Response

RE: EMBOJ-2024-116918R

Referee #1:

In this manuscript, Cho et al. investigate alternative splicing (AS) alterations within cancer driver genes. Their analysis uncovers AS alterations of PBRM1 exon 27 across various types of cancer tissues. Additionally, they present a regulatory mechanism of AS in PBRM1 and its potential implications for gene expression and clinical outcomes in cancer patients.

This manuscript is well-written and addresses an important topic, given the limited knowledge about functional differences among splicing isoforms in cancer tissues. Notably, this study highlights that differences in AS, even in the absence of differences in protein expression, can impact both gene expression and clinical outcomes. Consequently, this manuscript provides a new perspective on cancer by exploring AS alterations in depth.

Response

We would like to thank the reviewer for the detailed review and positive feedback on our manuscript. We believe that the revisions based on the comments of the reviewer significantly strengthened our manuscript.

Major concerns

- Figure 1: qPCR should be used to assess the AS pattern of PBRM1 E27 usage, normalized to gene expression, in human tissue samples.

- Additionally, the current approach appears insufficient to distinguish between E26 or E27 exclusion. Accordingly, figure 1E requires adjustment.

Response

We agree with the comments of the Reviewer about the insufficiency of our current approach

in identifying the *PBRM1* alternative splicing (AS) pattern in the human tissue samples. As suggested by the Reviewer, we performed a qPCR analysis using specific primers that targeted each *PBRM1* splice variants. Although *PBRM1* expression levels showed only slight differences in each tissue sample when normalized using reference gene expression levels, the large variation in cDNA concentrations between human tissue samples made it difficult to obtain clear results for identifying the alternative splicing pattern of *PBRM1*.

To address this issue, we developed a new method. We extracted DNA from the middle range bands of the PCR products, containing the amplified E26-included or E27-included *PBRM1* splicing variants. Next, we mixed the DNA samples with Apa1 restriction enzyme, which specifically cuts the middle region of *PBRM1* E27, followed by agarose gel electrophoresis. This approach allowed us to identify the proportion of the *PBRM1* splice variants more accurately. Our results indicate that *PBRM1* E27 alternative splicing patterns were significantly altered in cancer tissues compared to normal tissues, whereas the E26 alternative splicing patterns were not.

Accordingly, we have revised the Fig. 1E,F, and Fig. EV2B to present the exon usage quantification analysis of the *PBRM1* alternative exons. This revised data provides a clearer and more accurate depiction of the alternative splicing patterns of *PBRM1*, addressing the concerns raised by the reviewers.

- Figure 2: It is unclear how the current design of primers can distinguish transcripts with or without E26 or E27. Accordingly, the primer design should be clarified as primers should cover exon junctions to be specific.

Response

We apologize for the insufficient description of the design of the primer we used to analyze the *PBRM1* alternative splicing patterns. In the revised manuscript, we provided a detailed explanation of the primer design in the Results section (Page 5, line 124).

- The methods for quantification should be described. The changes observed on the gel do not appear as drastic as the quantification suggests.

Response

We included the additional description of the band intensity quantification method in the Methods section (Page 19, line 469) under “RT-PCR and qRT-PCR” of the revised manuscript to provide more detailed information.

The Reviewer also mentioned that the gel images and quantification results did not appear to match well. We think that this problem arose from the insufficient description of how the RT-PCR results were quantified and which gel bands were analyzed to determine the *PBRM1* splice variants. We believe that the revised manuscript, which includes additional information, will clarify these aspects.

Minor concerns

- The terminology used in Figure 2F and 2G is confusing. What exactly does "skipping isoform" refer to? Similarly, what does "E27+ isoform" signify?

Response

We thank the Reviewer for pointing out the confusion in our results. ‘Skipping isoform’ refers to a splice variant that does not include both E26 and E27. ‘E27+ isoform’ indicates the E27-included splice variant, whether E26 is included or excluded in. To avoid such confusion, we simplified the nomenclature to E26 (- or +) E27 (- or +) splice variants.

- Are transcripts with differential E27 exon usage associated with specific splice isoforms? RSEM analysis can be utilized to estimate isoform expression levels from RNA-seq data.

Response

As widely recognized in the field, Illumina-based RNA-seq faces several challenges in

accurately identifying and quantifying splice isoforms. These limitations stem from the short-read lengths, failing to capture full-length transcript structures and lack connectivity information between distant exons. Furthermore, biased coverage, particularly towards the 3' end, and difficulties in detecting low-abundance isoforms further complicate the bioinformatics analysis. The short-reads also make the distinction between similar isoforms and the identification of novel splice variants challenging.

Through RT-PCR analysis, we discovered that E27 can be incorporated into *PBRM1* transcripts independently of E26 inclusion in the manuscript. To address the comment of the Reviewer, we tried to capture full-length transcript structures with differential E27 usage by apply RNA-seq data. Analyzing the alternative splicing patterns of the *PBRM1* gene is particularly challenging due to its complex structure, comprising more than 30 exons and producing approximately 14 splice variants (Figure for the Reviewer). This complexity is compounded by the difficulties in distinguishing between similar isoforms and the uncertainty in assigning shared exon reads to specific variants. In addition, the uncertainty in assigning shared exon reads to specific isoforms leads to quantification inaccuracies. These constraints collectively hinder the comprehensive characterization of the full range of the splicing isoforms and their relative abundances.

Despite the limitations in capturing and quantifying full-length transcripts with differential E27 usage, we analyzed the expression levels of four distinct *PBRM1* splice variants (E26-E27-; E26+E27-; E26-E27+; E26+E27+; Figure for the Reviewer) using the TCGA TSVdb database (<http://www.tsvdb.com/index.html>). Our analysis revealed the higher prevalence of the E27-included splice variants in cancer tissues compared to normal tissues. In contrast, the E26-included splice variant expression patterns were similar to those lacking E26. These results we are consistent with our main conclusions.

Figure for the Reviewer. RSEM analysis of the *PBRM1* splice variants in UCEC.

- *eCLIP studies have been conducted, reporting RBFOX2 target genes. Is PBRM1 a known RBFOX2 target gene?*

Response

We would like to thank the Reviewer for the valuable information regarding the public RBFOX2 eCLIP data. We analyzed the eCLIP data from ENCODE and confirmed the RNA-binding peak of RBFOX2 protein in the upstream intronic region of *PBRM1* E27. We incorporated this result in Fig. 3A.

- *RBFOX2 is an important regulator of AS in human embryonic stem cells, which should be addressed in the discussion section.*

Response

As suggested by the Reviewer, RBFOX2 plays pivotal roles in human embryonic stem cells. Therefore the high-level conservation of the RBFOX binding sequences in the upstream intron of *PBRM1* E27 potentially indicates the importance of regulating *PBRM1* E27 alternative splicing for the embryonic stem cell maintenance. We expanded the description of this topic in the Discussion (Page 14, line 334) of the revised manuscript.

- Interestingly, PRBM1 E27, was significantly more included in cancer tissues than in normal tissues, whereas kidney renal cell carcinoma showed E27 exclusion. This finding should be addressed in the discussion section.

Response

We would like to thank the Reviewer for pointing out the need to address the different splicing patterns of *PBRM1* E27 in kidney renal cell carcinoma compared to other cancer types. We expanded the Discussion section (Page 16, line 387) to highlight this observation.

Referee #2:

Cho et al. describe a novel PBRM1 splicing isoform that impacts PD-L1 expression in tumors. This is a highly exciting and well-crafted manuscript. The authors go full circle, starting with the identification of splicing variants of cancer-relevant genes using genomic data. From there, they selected PBRM1 E27 for further analysis. They confirmed the alterations in cancer cells, determined that RBFOX2 regulates PBRM1 E27 splicing, and evaluated the downstream effects and clinical relevance. The findings are important since they provide a path to improve immunotherapy success. Regulation of PBRM1 splicing via AS oligos causes subsequent alterations in PD-L1 levels. I have just a few suggestions.

Response

We thank the Reviewer for the careful review of our manuscript and the positive feedback. We believe that the additional information in the revised manuscript will improve clarity.

- Are there any public CLIP datasets that show RBFOX2 binding to PBRM1?

Response

As the suggestion of the Reviewer, we found a public RBFOX2 eCLIP dataset and confirmed the binding of the RBFOX2 protein to the upstream intronic region of *PBRM1* E27. We included this result in Fig. 3A.

-A table with the results of the RNA seq analysis should be included in Sup data.

Response

We thank the Reviewer for highlighting this important point. We incorporated the FPKM values of each sample and the results of gene set enrichment analysis from the RNA-seq data in the Supplementary tables.

- The analysis shown in Fig 4b is very superficial. A more detailed GO, pathway, and network analysis should be performed, and a table with complete results should be included in the supplementary materials.

Response

We agree with the comment of the Reviewer concerning the limitations of the gene set enrichment analysis using total gene expression data. To address this, we performed a STRING protein-protein network analysis using the top 120 downregulated genes in Δ E27 HeLa cells and found that the most significantly enriched terms were associated with the interferon signaling pathway. We incorporated this result in Fig. EV5D.

Referee #3:

The authors present a nice study on the regulation of PBRM1 alternative splicing in cancer cells. Splicing factor RBFOX2 directly binds to PBRM1 intron, promoting Exon 27 skipping.

Exclusion of Exon 27 abolishes the binding of PBRM1 to PD-L1 promoters, resulting in reduced PD-L1 expression and enhanced response to immune therapy. Cancer cells exhibit significant alterations in splicing, generating potentially novel proteins for their benefit. Manipulating splicing has emerged as a promising therapeutic strategy for cancer treatment. This study exemplifies how immune checkpoint blockade therapy can be improved by modulating altered splicing in cancer cells. The analysis of splicing events appears to be well-performed and robust. Here are some suggestions could potentially improve the study:

Response

We would like to thank the Reviewer for the positive feedback on our manuscripts. We are convinced that the revisions based on the comments of the Reviewer strengthened our conclusions.

Major points:

1. The authors should elaborate more on the alternative splicing analysis of E26 and E27 in Figure 1, as the current description is unclear. Are E26 and E27 mutually exclusive or independent cassette exons? Additionally, the authors could design new pairs of primers, e.g., spanning E25/E26 or E25/27, to confirm rigorously that the upper band in Figure 1e represents the E27-containing isoform, similar to subsequent figures. Alternatively, the PCR product can be topo-cloned and Sanger sequenced for validation. While the PCR strategy in Figure 2 is sound, it is limited to two selected cell lines.

Response

We agree with the comment of the Reviewer about the insufficient conclusion of the PCR analysis in the cDNA array of the human tissues. As suggested by the Reviewer, we designed the new primer pairs targeting the exon junctions spanning E25/E26 and E25/E27 of the *PBRM1* transcripts and performed the qPCR analysis. However, due to differences in the cDNA concentration in each human tissue sample, we did not obtain clear results.

Therefore, we extracted DNA from all middle-range agarose gel bands containing

PCR products amplifying E26- or E27-included *PBRM1* splicing variants from human tissues. We then treated these samples with *Apa1* restriction enzyme, which specifically cuts the middle region of *PBRM1* E27, followed by agarose gel electrophoresis to identify the proportion of the E26- and E27-included *PBRM1* splicing variants. We observed that the E27 inclusion significantly increased in the uterine and breast cancer tissues compared to normal tissues, while E26 inclusion was not significantly altered. We incorporated this result in Fig. 1E,F, as well as Fig. EV2B to present the *PBRM1* alternative splicing patterns in human tissues.

The Reviewer also commented that the results suggesting RBFOX2-mediated regulation of *PBRM1* alternative splicing are limited to two cell lines. To address this aspect, as an additional experiment, we overexpressed RBFOX2 in HEK293 and MCF7 cells and analyzed the *PBRM1* alternative splicing patterns, confirming that RBFOX2 protein induces E27 exclusion in *PBRM1*. We incorporated these results in Fig. EV4C,D.

2. In Figure 3, while minigene reporters are valuable tools for studying alternative splicing, they may not fully reflect endogenous regulatory mechanisms. It is suggested to use CRISPR to modify/mutate the RBFOX2 binding motifs and probe splicing changes, which would strengthen the mechanistic study. The authors should clarify how they determined the mutational sequences as ACCTAC and explore if other mutations yield similar effects. Are there any RBFOX2 binding motifs downstream of E27 that might also influence E27 inclusion?

Response

We would like to thank the Reviewer for the valuable suggestion. We considered CRISPR-Cas9-mediated modification of the RBFOX2 binding motifs but found it particularly challenging due to the lack of the PAM sequences around sites 1 and 2 as well as the proximity of the consensus sequence for mRNA splicing to site 1.

Although we did not modify the RBFOX binding motifs to confirm the endogenous regulatory mechanism of RBFOX2 for *PBRM1* E27 alternative splicing, we incorporated the

RBFOX2 eCLIP-seq result in Fig. 3A, confirming the RBFOX2 binding peaks on the endogenous intronic region upstream of *PBRM1* E27. We believe this result indicates that the RBFOX2 protein is endogenously involved in the regulation of *PBRM1* E27 alternative splicing.

Moreover, we would like to thank the Reviewer for your insightful comment regarding the results of the minigene reporter analysis. We determined the mutational sequences as ACCTAC based on these sequences are not known targets of splicing factors from previous studies. To explore whether other mutations yield similar effects, we conducted additional experiments using the minigene constructs with transversion mutations at the RBFOX2 binding sites. These experiments confirmed that other mutations of the RBFOX2 binding motif consistently disrupted the regulatory mechanism of *PBRM1* E27 AS. We included these results in Fig. EV4E-G.

3. Can the authors validate the mechanisms in multiple cancer cell lines:

- *Overexpression of RBFOX2 in RBFOX2-low cell lines promotes PBRM1 Exon 27 skipping;*
- *knockdown of E27-containing isoforms in tumor cells reduces PD-L1 expression.*

Response

We would like to thank the Reviewer for the valuable suggestion. To validate our results using multiple cancer cell lines, we conducted additional experiments. We overexpressed RBFOX2 in HEK293 and MCF7 cells and confirmed the exclusion of E27 from *PBRM1* in these cells. Furthermore, we designed a *PBRM1* E27-specific siRNA and transfected it into WT and Δ E27 HeLa as well as the MDA-MB-231 cells, the latter of which positively expressing PD-L1 and being commonly used for studying the regulation of PD-L1 expression. These results revealed that knockdown of E27-included PBRM1 repressed PD-L1 expression in the HeLa and MDA-MB-231 cells. These findings support the role of RBFOX2 in promoting *PBRM1* E27 exclusion and demonstrate that the reduction of E27-included isoforms in tumor cells can lead to decreased PD-L1 expression. We included these results to Fig. EV5I,J.

4. In Figure 5D, the enrichment of PD-L1 binding does not significantly differ between negative control regions and PD-L1-negative control regions (%input), raising concerns about the specificity of this binding. Are there previous datasets available to corroborate the binding of PBRM1 to the PD-L1 promoter? Is PBRM1 a direct DNA-binding factor with known DNA motifs, and if so, do they exist in the PD-L1 promoter? In addition to qPCR, the authors should present gel plots of ChIP-amplified regions, showing a single specific band with differences in intensity as supporting evidence, similar to what they did in Figure 3b RIP experiment. Luciferase reporter assays with the PD-L1 promoter could also demonstrate the functions of potential binding.

Response

We would like to thank the Reviewer for your insightful comments regarding the PBRM1 splicing isoform binding to the *PD-L1* promoter region. We agree with the concerns of the Reviewer about the PBRM1 binding specificity. PBAF complexes, including PBRM1, function as chromatin remodelers, potentially yielding a relatively weak binding specificity. This might explain why the qRT-PCR-analyzed PBRM1 and ARID2 binding to the *PD-L1* promoter region displayed a relatively small difference compared to those of the negative control region.

The Reviewer also raised a question about the binding mechanism of PBRM1 to the *PD-L1* promoter regions. The detailed underlying mechanism of PBRM1 binding specificity to regulate target genes remains elusive. We attempted to confirm whether PBRM1 protein directly binds to the *PD-L1* promoter region using an *in vitro* transcription and translation system, but encountered challenges due to the protein's large size and the presence of intrinsically disordered regions. Nonetheless, based on previous studies, we expected that the DNA binding activity of the HMG domain, encoded in E27 of the *PBRM1*, and the acetylated histone binding activity of the PBAF complex might be important for binding to the *PD-L1* promoter. Furthermore, analysis of ENCODE data confirmed the presence of ARID2 and H3K27ac protein peaks in the *PD-L1* promoter region, prompting us to select this region for ChIP assay (Figure for the Reviewer A). For the ChIP-qPCR experiments, we validated the specificity of primers through qPCR using human chromosomal DNA, followed by melting curve analysis and agarose gel electrophoresis. The melting curves from ChIP-qPCR analysis using anti-PBRM1 or anti-ARID2 in WT and Δ E27 cells showed single specific peaks at the

same temperature as the peak of the melting curve from qPCR using human chromosomal DNA (Figure for reviewer B). This consistency in melting curve profiles confirms the specificity and reliability of our ChIP-qPCR results.

As suggested by the Reviewer, we constructed the *PD-L1* promoter luciferase vector and performed dual-luciferase assay in WT and $\Delta E27$ cells to confirm the *PD-L1* promoter activity. Our results revealed that the *PD-L1* promoter activities significantly decreased in $\Delta E27$ cells compared to WT cells. However, the difference in *PD-L1* promoter activities between WT and $\Delta E27$ cells was markedly smaller than the difference in endogenous *PD-L1* mRNA levels. We incorporated this result in Fig. EV5G. Overall, we speculate that E27-included PBRM1 may regulate *PD-L1* expression through epigenetic mechanisms in corroboration with the PBAF complex rather than by acting directly as a transcription factor. We expanded the description of this topic in the Results section (Page 9, line 210) of the revised manuscript.

Figure for the Reviewer. (A) ChIP-seq peaks for ARID2 and H3K27ac protein on the *PD-L1* gene. (B) Melting curve analysis of ChIP-qPCR products amplifying the *PD-L1* promoter region in the indicating samples.

Minor points:

1. In Figure 4d, why are there no error bars in the RNA-seq data?

Response

We originally presented the graph using only the average FPKM values of the RNA-seq data. However, we revised this result to include the replicate data with error bars for better representation.

2. How many PBRM1 isoforms exist in terms of Exon 26 and 27 inclusion/skipping? Does the $\Delta E27$ form in the engineered HeLa cell line contain E26 or not?

Response

The *PBRM1* gene contains approximately 30 exons. According to the TCGA SpliceSeq database, there are 10 alternative splicing events profiled in *PBRM1* transcripts. With different splice combinations, nine transcripts of *PBRM1* are displayed in GENECODE version 46. However, the analysis of percent spliced in index (PSI) values for each splicing event confirmed that, except for the alternative splicing of E26 or E27, other splicing events rarely occur in *PBRM1* transcripts around 0.5 to 10%. We speculate that these splicing events may occur by chance due to misprocessed RNA splicing. Consequently, the four isoforms analyzed in this study represent the predominant variants of *PBRM1*.

We apologize for the insufficient description of the $\Delta E27$ cell lines. We established these cell lines using the CRISPR-Cas9 system targeting the 3' splice site of *PBRM1* E27. We confirmed that these cell lines exhibited depleted E27 inclusion, while the inclusion rates of E26 remained unchanged, as shown by our RT-PCR analysis (Fig. 4A). For better clarity, we described additional information in the Results section (Page 8, line 180) of the revised manuscript.

3. What are the PD-L1 expression levels and responses to PD-1 blockade therapy in KIRC?

Does reduced inclusion of E27 correlate with a better response to immune therapy?

Response

We would like to thank the Reviewer for the insightful comments. As suggested by the reviewer, we analyze the impact of *PD-L1* expression status on the overall survival of patients with ccRCC after PD-1 blockade therapy. The results showed that, although there is a higher proportion of patients with high *PD-L1* expression levels in the *PBRM1* WT and low *RBFOX2* mRNA levels group, *PD-L1* expression status did not significantly affect resistance to PD-1 blockade therapy. This suggests that the impacts of *PBRM1* alternative splicing on clinical outcome of PD-1 blockade therapy may be influenced by variable factors other than just regulating *PD-L1* expression levels. We included these results in Fig. EV7G,H and described these findings in the Result section (Page 13, line 303) of the revised manuscript.

4. For most qPCR quantifications in the paper, the data points in all control bars are consistently the same without variation. The authors should explain how the quantification was performed.

Response

We thank the Reviewer for highlighting this important point. We performed independent measurements on distinct biological samples to increase the reliability of the experimental results. We quantified the qRT-PCR analysis by calculating the fold change in the mRNA expression levels relative to the negative control, which was set to one. The data points in the control bars consistently present a value of one because we merged the repeated qRT-PCR results, standardizing the control to this baseline value. We revised the Methods section (Page 26, line 631) under ‘Statistical Analysis’ in the revised manuscript to include this information.

5. In the analysis of KIRC TCGA data, how many mutation types are included in the PBRM1 Mut group? Are all of them loss-of-function mutations?"

Response

We would like to thank the Reviewer for pointing out what we missed. The *PBRM1* Mut group includes stop gain, splice, missense, and inframe mutations, the majority being stop gain mutations. We incorporated the result presenting the detailed mutation frequencies in Fig. EV7A.

Dear Dr Kim,

Thank you for submitting your revised manuscript (EMBOJ-2024-116918R) to The EMBO Journal, as well as for your patience with our response at this time of the year. Your amended study was sent back to the referees for their re-evaluation, and we have received comments from all of them, which I enclose below. As you will see, the experts stated that the work has been substantially improved by the revisions and they are now in favour of publication, pending minor revision.

Thus, we are pleased to inform you that your manuscript has been accepted in principle for publication in The EMBO Journal.

Please consider the remaining minor point of referee #1 carefully and amend the manuscript accordingly.

We also now need you to take care of a number of minor issues related to formatting and data presentation as detailed below, which should be addressed at re-submission.

Please contact me at any time if you have additional questions related to below points.

As you might have seen on our web page, every paper at the EMBO Journal now includes a 'Synopsis', displayed on the html and freely accessible to all readers. The synopsis includes a 'model' figure as well as 2-5 one-short-sentence bullet points that summarize the article. I would appreciate if you could provide this figure and the bullet points.

Thank you for giving us the chance to consider your manuscript for The EMBO Journal.
I look forward to your final revision.

Again, please contact me at any time if you need any help or have further questions.

Kind regards,

Daniel Klimmeck

>> Author Contributions: Please remove the author contributions information from the manuscript text. Note that CRediT has replaced the traditional author contributions section as of now because it offers a systematic machine-readable author contributions format that allows for more effective research assessment. and use the free text boxes beneath each contributing author's name to add specific details on the author's contribution.

More information is available in our guide to authors.
<https://www.embopress.org/page/journal/14602075/authorguide>

>> Authors: All corresponding authors need institutional email addresses entered into our online system and accounts need the ORCID number linked (E-M.K.). Please see below for additional information.

>> Please provide a completed Author Checklist.

>> Callouts: please add a callout to Fig. 6J in the running text.

>> Add a Reagents and Tools table to the Methods section, listing key reagents, experimental models, software and relevant equipment.

>> Dataset EV legends: 10 data sets are uploaded together in one file. The file should be split and each dataset should be uploaded individually. Datasets 1 and 10 should be renamed Table EV1 and Table EV2, the other files should be uploaded as Dataset EV1, Dataset EV2, etc. .

>> Appendix file: There are currently eight EV figures. Please compile three figures as an appendix: a PDF with the figures and their legends, named "Appendix Figure S1" etc. The appendix will need a table of contents with page numbers added on the first page. Adjust the figure callouts to the new nomenclature.

>> Data availability section: please name all individual SRA datasets individually and provide hyperlinks.

>> Provide an animal welfare statement in the Methods section.

>> Source Data: please upload as one (zipped) file per figure.

>> Consider additional changes and comments from our production team as indicated below:

- DAS:

Please note that the specific URLs for SRX20520582-SRX20520585 under BioProject PRJNA976477 datasets are not provided in the data availability statement.

- Figure legends:

1. Please note that the exact p values are not provided in the legends of figures 1d-f; 2c, f-g; 4c-f; 5d; 6c-f, h; EV 3c; EV 4c-d; EV 5c, e-g, i-j.
2. Please indicate the statistical test used for data analysis in the legends of figures 2c, f-g; 4c; 6h-i; EV 5c.
3. Please note that the box plots need to be defined in terms of minima, maxima, centre, bounds of box and whiskers, and percentile in the legends of figures 1d; 6c-d; EV 2a.
4. Please note that information related to n is missing in the legends of figures 1d; EV 2a; EV 4c-d.
5. Please note that n=2 in figures 5d-e.
6. Although 'n' is provided, please describe the nature of entity for 'n' in the legends of figures 2f-g; 4d-f; 5d-e; 6c-f; EV 3c; EV 4a-b; EV 5e-f, i-j.
7. Please note that the error bars are not defined in the legends of figures 2f-g; 5d-e; EV 4a-d; EV 5e-f, i-j.

Please note that as of January 2016, our new EMBO Press policy asks for corresponding authors to link to their ORCID iDs. You can read about the change under "Authorship Guidelines" in the Guide to Authors here: <http://emboj.embopress.org/authorguide>

In order to link your ORCID iD to your account in our manuscript tracking system, please do the following:

1. Click the 'Modify Profile' link at the bottom of your homepage in our system.
2. On the next page you will see a box half-way down the page titled ORCID*. Below this box is red text reading 'To Register/Link to ORCID, click here'. Please follow that link: you will be taken to ORCID where you can log in to your account (or create an account if you don't have one)
3. You will then be asked to authorise Wiley to access your ORCID information. Once you have approved the linking, you will be brought back to our manuscript system.

We regret that we cannot do this linking on your behalf for security reasons. We also cannot add your ORCID iD number manually to our system because there is no way for us to authenticate this iD number with ORCID.

Thank you very much in advance.

Please use the link below to submit your revision:

Referee #1:

This manuscript presents important data, given the limited knowledge about functional differences among splicing isoforms in cancer tissues. Since the authors were unable to use qPCR to quantify the splice isoforms of interest, they developed an innovative approach to overcome this challenge. For validation, all RT-PCR bands should be sequenced to confirm that the annotated variants match the schematics shown. Other than that, I have no further comments.

Referee #2:

I was very excited with the first version of this article and had only minor comments. The authors did a great job addressing my suggestions and other reviewers' concerns. This is an excellent article that brings new and exciting discoveries and therefore, it merits publication in EMBOJ.

Referee #3:

The authors have satisfactorily addressed my major concerns in the earlier version of the manuscript. Multiple new experiments have been included to support their data. For technical challenging experiments, they authors actively tried alternative approaches to solve my concerns. I am in favor of publishing the revised paper and congratulations to the authors on finishing this nice study.

The point-by-point response to the editor comments.

We sincerely thank the Editor for the constructive comments. We have carefully addressed all the editorial issues. We believe our revised manuscript now presents a clearer and more accurate conclusion.

The point-by-point response to the referee comments.

Referee #1:

This manuscript presents important data, given the limited knowledge about functional differences among splicing isoforms in cancer tissues. Since the authors were unable to use qPCR to quantify the splice isoforms of interest, they developed an innovative approach to overcome this challenge. For validation, all RT-PCR bands should be sequenced to confirm that the annotated variants match the schematics shown. Other than that, I have no further comments.

Response

We thank the reviewer for the positive feedback on our approach to addressing the challenge of quantifying the *PBRM1* splice variants using qPCR analysis from the tissue cDNA array. In response to the reviewer's comments, we validated the nucleotide sequence of the PCR products and confirmed that they are indeed amplified products of *PBRM1* splice variants. We have incorporated this result in Appendix Fig. 2B.

Referee #2:

I was very excited with the first version of this article and had only minor comments. The

authors did a great job addressing my suggestions and other reviewers' concerns. This is an excellent article that brings new and exciting discoveries and therefore, it merits publication in EMBOJ.

Response

We appreciate the reviewer's for recognizing the significance in our findings and supporting our manuscript's publication in *the EMBO journal*.

Referee #3:

The authors have satisfactorily addressed my major concerns in the earlier version of the manuscript. Multiple new experiments have been included to support their data. For technical challenging experiments, they authors actively tried alternative approaches to solve my concerns. I am in favor of publishing the revised paper and congratulations to the authors on finishing this nice study.

Response

We greatly appreciate the reviewer's positive feedback. We believe that the additional experiments conducted in response to the reviewer's comments have further strengthened our manuscript.

Dear Dr Kim,

Thank you for submitting the revised version of your manuscript. I have now evaluated your amended manuscript and concluded that the remaining minor concerns have been sufficiently addressed.

I am thus pleased to inform you that your manuscript has been accepted for publication in the EMBO Journal.

Related, I kindly ask for your consent on keeping the referee figures included in this file.

On a different note, I would like to alert you that EMBO Press offers a format for a video-synopsis of work published with us, which essentially is a short, author-generated film explaining the core findings in hand drawings, and, as we believe, can be very useful to increase visibility of the work. Please see the following link for representative examples and their integration into the article web page:

<https://www.embopress.org/doi/full/10.15252/emj.2019103932>

Best regards,

Daniel Klimmeck

Daniel Klimmeck, PhD
Senior Editor
The EMBO Journal
EMBO
Postfach 1022-40
Meyerhofstrasse 1
D-69117 Heidelberg
contact@embojournal.org
Submit at: <http://emboj.msubmit.net>
